# Structural basis of denuded glycan recognition by SPOR domains in bacterial cell division

Martín Alcorlo [1], David A. Dik[2], Stefania De Benedetti[2], Kiran V. Mahasenan [2], Mijoon Lee[2], Teresa Domínguez-Gil[1], Dusan Hesek[2], Elena Lastochkin[2], Daniel López [3], Bill Boggess[2], Shahriar Mobashery[2]* & Juan A. Hermoso[1]*

SPOR domains are widely present in bacterial proteins that recognize cell-wall peptidoglycan strands stripped of the peptide stems. This type of peptidoglycan is enriched in the septal ring as a product of catalysis by cell-wall amidases that participate in the separation of daughter cells during cell division. Here, we document binding of synthetic denuded glycan ligands to the SPOR domain of the lytic transglycosylase RlpA from *Pseudomonas aeruginosa* (SPOR-RlpA) by mass spectrometry and structural analyses, and demonstrate that indeed the presence of peptide stems in the peptidoglycan abrogates binding. The crystal structures of the SPOR domain, in the apo state and in complex with different synthetic glycan ligands, provide insights into the molecular basis for recognition and delineate a conserved pattern in other SPOR domains. The biological and structural observations presented here are followed up by molecular-dynamics simulations and by exploration of the effect on binding of distinct peptidoglycan modifications.

[1] Department of Crystallography and Structural Biology, Instituto de Química-Física "Rocasolano", Consejo Superior de Investigaciones Científicas, Madrid, Spain. [2] Department of Chemistry and Biochemistry, University of Notre Dame, Notre Dame, Indiana 46556, USA. [3] National Centre for Biotechnology, Spanish National Research Council (CNB-CSIC), 28049 Madrid, Spain. *email: mobashery@nd.edu; xjuan@iqfr.csic.es

Peptidoglycan (PG), the major constituent of the cell wall, is simultaneously biosynthesized and degraded during a complex set of events in bacterial division. These entail the formation of the septum and invagination of the membranes, culminating in daughter-cell separation. As the largest bacterial macromolecule, the cell wall is a single crosslinked entity that encases the entire bacterium. The general structure of the PG is that of a repeating backbone of N-acetylglucosamine (NAG) and N-acetylmuramic acid (NAM), connected by β-1,4-glycosidic linkages. A unique bacterial peptide stem, typically L-Ala-γ-D-Glu-m-DAP-D-Ala-D-Ala in Gram-negative bacteria, is appended to the NAM unit (Fig. 1a). The assembly of the PG and the crosslinking of its peptide stems at the septum is mediated by a multiprotein complex called the divisome, which assembles at the middle of the replicating cell[1]. Divisome assembly is primarily driven by a network of protein-protein interactions, whereby a new division protein is recruited by binding to other divisome proteins[2–4]. Structural details of the chemical modifications of the septal PG in the course of these events have not been fully elucidated. However, it is known that the sporulation-related repeat (SPOR) domains (Pfam 05036), ubiquitous in prokaryotes, play a central role in directing proteins to the septum by recognition of the unique septal PG (rather than recognition by other division proteins), which is enriched in denuded glycans, the PG devoid of peptide stems[5]. These denuded glycans are transiently formed at the bacterial division site by the action of periplasmic amidases[6–11], which cleave septal PG to allow daughter-cell separation (Fig. 1b, c). Sequence analysis reveals 12745 proteins from over 4513 species have the SPOR domain, spanning both Gram-negative and Gram-positive bacteria (as of November 2019). All known proteins possessing SPOR domains are either involved in cell division or morphogenesis. For example, four proteins with SPOR domains have been identified in Escherichia coli (DamX, DedD, FtsN, and RlpA), all of which are involved in cell division,

with one (FtsN) assessed as indispensable[12,13]. Bacillus subtilis and Pseudomonas aeruginosa, as two other examples, have three and five proteins with SPOR domains, respectively. SPOR domains are typically comprised of approximately 70 amino acids, whose sequences are not highly conserved at < 20% sequence identity in pairwise alignments. The majority of SPOR-domain-containing proteins are predicted to have this small domain as a folded unit coupled to an N-terminal transmembrane anchor through disordered and low-complexity regions. However, SPOR domains are also found in association with a variety of other modules/domains, the vast majority of which are associated with cell division[14].

Binding of PG by SPOR domains was documented first by the Vollmer laboratory[15]. Although not conserved at the level of primary structure, SPOR domains exhibit similar three-dimensional core architectures related to the ribonucleoprotein (RNP)-fold superfamily[16–18]. The core fold is an antiparallel β-sheet flanked on one side by two helices. Previous structure-function studies, especially the ones based on site-directed mutagenesis of SPOR-DamX and SPOR-FtsN[5,19,20], revealed that residues important for septal localization and PG-binding are in the exposed face of the β−sheet. Notwithstanding the availability of NMR structures for the SPOR domains of FtsN[21] and DamX from E. coli[20], and of the sporulation protein CwlC of B. subtilis[22], none sheds light on the unique recognition of the denuded glycan, as we will discuss.

In the present report, we disclose the X-ray structure of the SPOR domain of rare lipoprotein A (RlpA) from P. aeruginosa (SPOR-RlpA). RlpA is a lytic transglycosylase (LT)[23] predicted to be anchored to the inner leaflet of the outer-membrane[24,25] (Fig. 1c). LTs are bacterial enzymes that cleave the β-1,4-glycosidic bond between sequential NAM and NAG, which are involved in cell-wall biosynthesis and recycling, cell division, insertion of cell-wall structures and cell-wall antibiotic

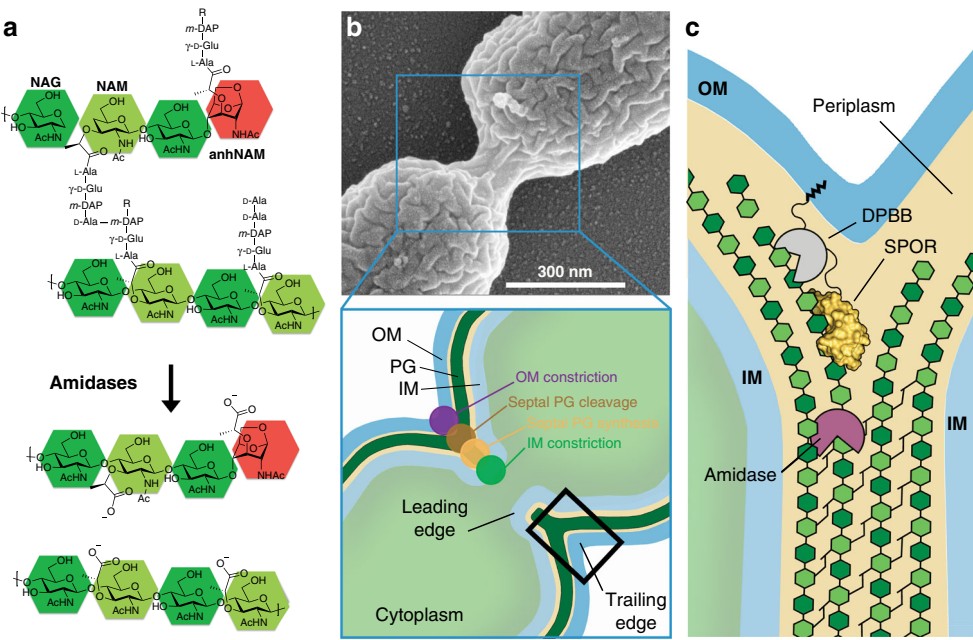

**Fig. 1** Cell envelope constriction in Gram-negative bacteria. **a** Components of PG in Gram-negative bacteria and production of denuded glycan in septal PG by amidases. NAM is highlighted in dark green, NAG in light green and 1,6-anhydroNAM (anhNAM) in red. **b** Top, scanning-electron micrograph showing advanced cytokinesis between P. aeruginosa PAO1 daughter cells. Down, schematic cartoon adapted from Gray et al.[30] representing the cell-division septum and coordinated processes involved in cell-envelope constriction. **c** Close-up diagram of the division site (corresponding to the black boxed area in (**b**), according to the model first put forth by Jorgenson et al.[23], showing the coordinated action of peptidoglycan amidases (purple) and the glycolytic activity of RlpA upon recognition of the naked glycan chains by its SPOR domain (in yellow surface). OM outer membrane, IM inner membrane. The catalytic DPBB domain of RlpA is colored in gray. The protein has a covalently attached lipid at the amino terminus.

detection[26]. Of all SPOR-domain-containing proteins, RlpA is the most highly conserved across bacterial species[23], which underscores its important physiological role in bacteria. It is predicted to have two domains, a catalytic "double-ψ β-barrel" domain (DPBB; Pfam 03330) and a C-terminal SPOR domain. RlpA septal ring localization has been shown in *E. coli*[12,13] and in *P. aeruginosa*[23] and its function is needed for efficient separation of daughter cells and maintenance of rod shape in *P. aeruginosa*[23] but not in *E. coli*[12,13]. RlpA exhibits a requirement for denuded glycan strands and a regulatory mechanism has been proposed in which amidases and RlpA work in tandem to degrade PG at the septum in division[23,27,28] (Fig. 1c). In addition, it has been recently shown that RlpA of *E. coli* directly interacts with the essential cell-division protein FtsK[29]. By the use of authentic synthetic PG fragments, we hereby provide evidence in a series of mass spectrometric and crystallographic experiments, and molecular dynamics simulations, that the SPOR-RlpA domain indeed recognizes denuded PG. Moreover, we provide a structural context for this recognition event. This structural foray into the recognition of the denuded PG by the SPOR domain has broad implications for understanding of bacterial sequestration of the PG during the cell-division process.

## Results

**Binding of PG derivatives to the SPOR domain.** The gene for the SPOR-RlpA domain (residues 264–342 of RlpA) of *P. aeruginosa* was cloned and overexpressed in *E. coli* and the protein was purified to homogeneity and different synthetic PG fragments (Fig. 2) were produced (Supplementary Table 1, see Materials and Methods). The MS analysis showed the SPOR domain at 11,117.4 Da (Fig. 3a), corresponding to the predicted mass of the intact monomer. We next assessed binding of five distinct synthetic PG derivatives (Fig. 2a). These were compounds **1**, **2**, **3**, **4** and **5**, which were prepared for this study by multistep syntheses (see Materials and Methods). For compound **1** a binding event is observed at 12,106.8 Da, corresponding to a predicted mass shift of 989.3 Da for the ligand (Fig. 3b). Similarly, for compound **2** and **3** binding takes place with molecular masses of 12,104.2 and 12,074.7 Da, respectively (Fig. 3c). The binding interaction of a denuded PG octasaccharide **4** was also assessed. Again, a binding event occurred at 13,063.3 Da, corresponding to the predicted mass shift for the binding of the octasaccharide to the domain (Fig. 3d). Next the binding ability of compound **5** was assessed, which includes the full-length peptide stems. No binding was detected (Fig. 3e), affirming that SPOR-RlpA binds exclusively PG devoid of peptide stems. Although we did not vary concentrations of the ligands to

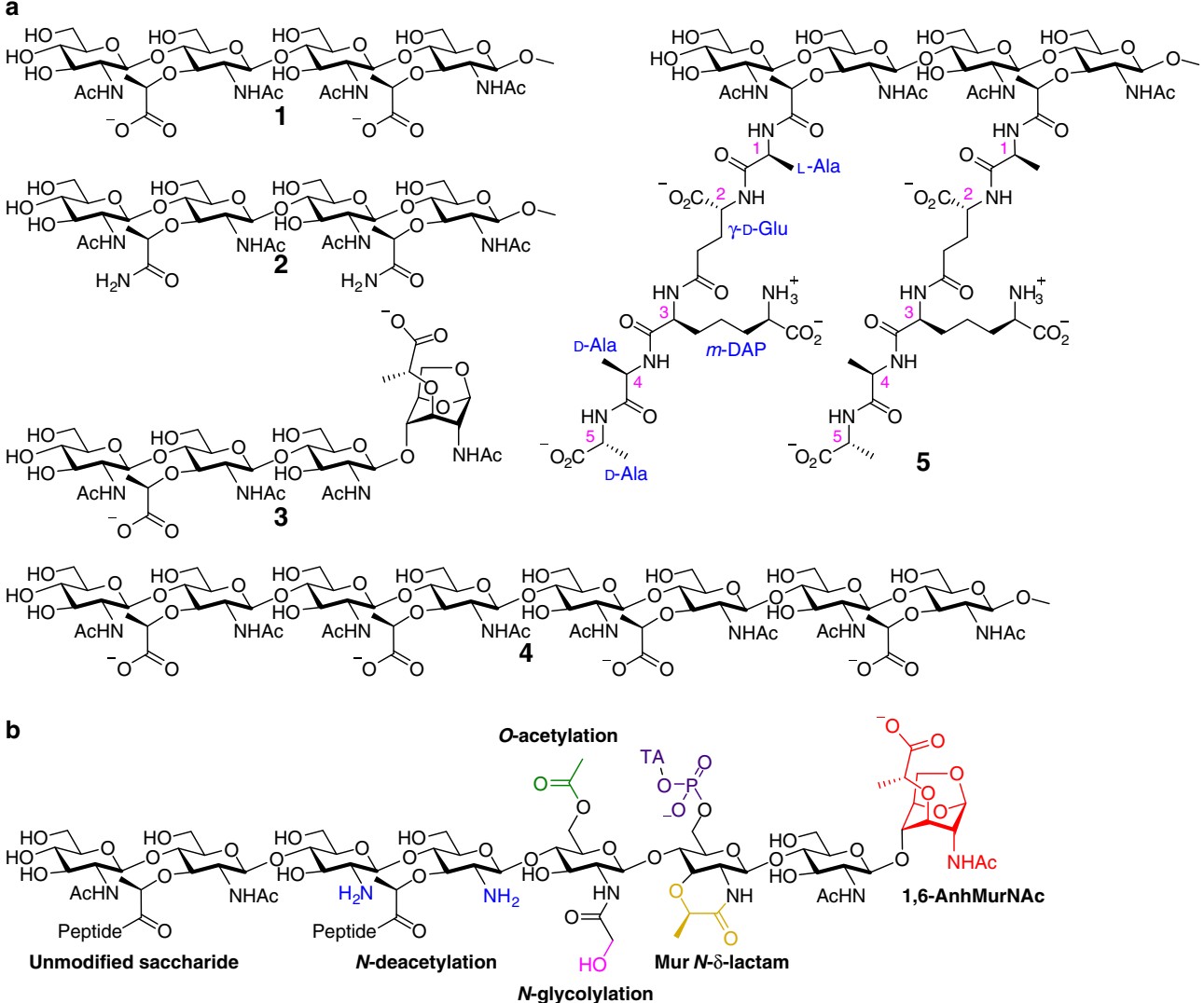

**Fig. 2** PG derivatives and modifications. **a** Chemical structure of the authentic synthetic PG-based compounds used in this study. **b** Variations in the peptidoglycan structure. TA stands for teichoic acid (in Gram + bacteria).

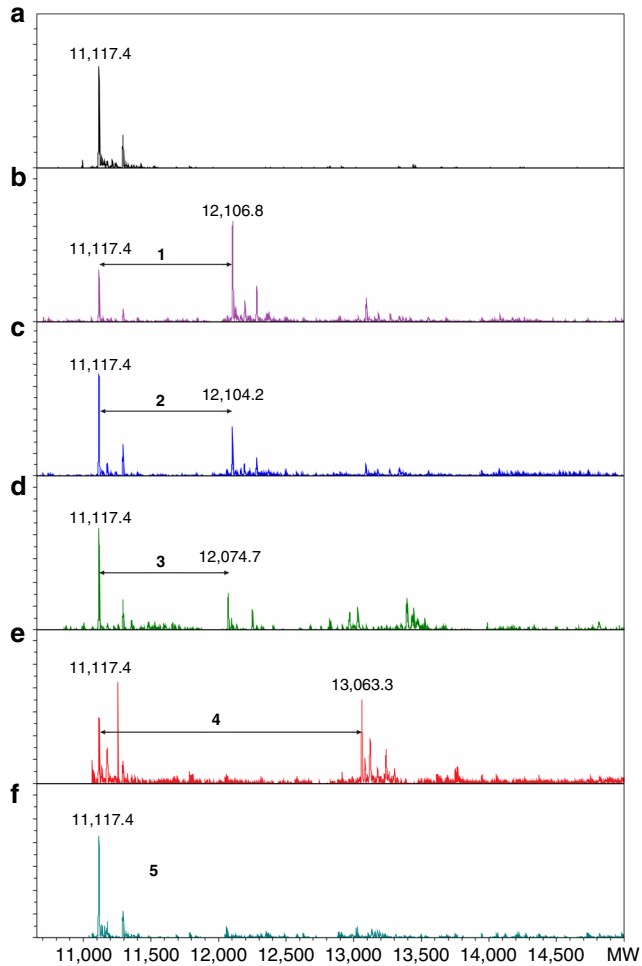

**Fig. 3** SPOR-RlpA binding to PG derivatives. Representative extracted-ion chromatograms of the non-denaturing mass spectrometry results displaying the SPOR-RlpA domain as an intact monomer and in complex with distinct authentic synthetic muropeptides. **a** SPOR-RlpA domain alone (20 μM) and the 20 μM domain plus 500 μM of (**b**) **1**, (**c**) **2**, (**d**) **3**, (**e**) **4**, and (**f**) **5**. The intensity of the signal (*Y* axis) is plotted vs Mw (average molecular weight) of the indicated species. Masses for the different compounds are as follows: 988.4 Da (compound **1**), 986.4 (compound **2**), 956.4 (compound **3**), 1944.7 (compound **4**) and 2016.9 (compound **5**).

evaluate the dissociation constants for them because of their limited supply, these data provide a qualitative assessment of the affinity of the protein to the ligands, which are as follows: **1** > **4** > **2** > **3**. The ligand **2**, the amide variant of the lactate moiety of the denuded PG, is not found in nature. It was prepared for this study to explore the role of the negative charge on the lactate in its interactions with the SPOR domain. As observed in the crystal structures (vide infra) the charged carboxylate makes stronger electrostatic interactions with the domain surface, and thus ligand **1** binds the protein more avidly than **2**. It is also of interest to note that compound **1** binds the protein better than **3**. This observation suggests that the SPOR domain directs RlpA to the core PG for its preferential endolytic reaction, as opposed to the "peripheral" PG that end in anhNAM (as in **3**). Furthermore, the octasaccharide compound **4** binds to a single SPOR domain (as opposed to two or more). This argues that the SPOR domain binds likely to the central four saccharides of the longer PG with shorter stubs protruding from the ligand-binding site of the protein. It has been shown in vivo previously that the physiologically relevant binding site for SPOR domains is a region of glycan strand that lacks stem peptides[5]. However, our

experiments provide, for the first time using a homogeneous chemically defined ligand, direct evidence that a SPOR domain targets naked PG chains.

**Crystal structure of the SPOR domain of RlpA**. The crystal structure of the SPOR-RlpA was solved at atomic resolution (1.2 Å) by ab initio phasing using the program arcimboldo[31] (Supplementary Table 2, Supplementary Fig. 1, Materials and Methods). The structure provided superb electron density for amino acids 265–342 (Supplementary Fig. 2A) of RlpA. The domain assembles into a concave four-stranded antiparallel β-sheet, complemented on one side by two α-helices (Fig. 4a). The structure resembles an open right-hand palm (the four finger tips being the β2-β3 turns in Fig. 4) with maximum dimensions of 35 × 28 × 14 Å.

The main frame of the fold consists of a curled β-sheet. The β1 strand forms the central part of the fold with the adjacent β3 arranged in an antiparallel manner, and β4 and β2 contacting β1 and β3, respectively. As in other SPOR domains (Supplementary Fig. 3), the SPOR-RlpA domain is assembled by two repeats. The first repeat (residues 264–303) comprises β1, α1, and β2, while the second repeat (residues 304–342) comprises β3, α2, and β4. Numerous contacts are found between the two repeats. In addition to the hydrogen-bonding network between the β-strands, hydrophobic interactions play an essential role in linking the two repeats in a single compact structure (Supplementary Fig. 4).

**Crystal structure of SPOR domain in complex with PG derivatives**. In order to characterize how the SPOR-RlpA domain interacts with denuded PG, we solved the X-ray structure of the complex with compound **1** (Fig. 2a). Co-crystallization and soaking were both performed in parallel and resulted in two structures for the SPOR-RlpA:**1** complex at near-atomic resolution (1.30 and 1.48 Å, respectively) (Supplementary Table 2, see Materials and Methods). The two structures are identical (rmsd of 0.16 Å for the 77 Cα atoms); the description provided herein is for the 1.30 Å resolution cocrystallization complex. The electron-density for compound **1** bound to the SPOR domain is excellent (Supplementary Fig. 2B). The PG-binding site in the SPOR domain is located within the concave face of the β-sheet (the "palm" side in the top depiction of Fig. 4a), presenting a short funnel-like cavity (15 Å long by a width raging from 8–18 Å) (Fig. 4b) in which the tetrasaccharide is ensconced in an extended conformation. The opposite face (convex face) of the SPOR domain does not present any of the features observed for compound **1** interaction (Supplementary Fig. 5), thus pointing to a single PG-binding site per SPOR domain, consistent with the mass spectrometric data. This is also consistent with previous mutagenesis studies of FtsN[19], in which essentially every surface exposed amino acid on this side of the domain was altered without finding any noteworthy effects on septal localization, therefore strengthen our argument that the only glycan binding site is the β−sheet. The positions of the sugar units have been labeled 1 to 4, corresponding to NAM(1), NAG(2), NAM(3), NAG(4). The high-resolution data for the SPOR-RlpA:**1** complex allowed us to map all the amino acids and water molecules involved in glycan-chain recognition (Fig. 4c and Supplementary Fig. 6). The following observations are made for this complex. (i) There is no discernable conformational change in the backbone of the SPOR domain upon ligand binding (rmsd of 0.27 Å for Cα superimposition of SPOR domain alone and in complex with **1**, Supplementary Fig. 7). Therefore, our results do not support the existence of "open" and "closed" conformations as a general mechanism in SPOR domain recognition, as previously proposed[20]. Changes only

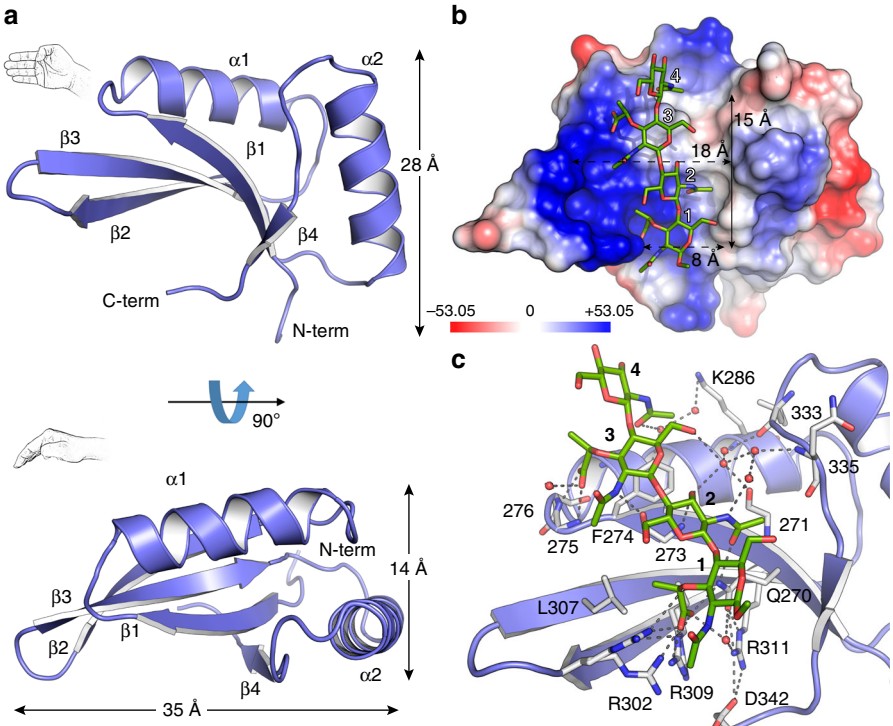

**Fig. 4** Crystal structure of the SPOR-RlpA domain and PG binding. **a** The ribbon structure is displayed in two orientations at 90° of each other, with the assigned secondary structure elements depicted. **b** The electrostatic potential of the surface of the domain is shown in the SPOR:**1** complex. Compound **1** is represented as capped sticks colored by atom types (carbons in green, oxygens in red and nitrogens in blue). The color key shows the Poisson-Boltzmann electrostatic-potential surface (color bar range ± 53.05 kT/e). **c** The details of the recognition of denuded PG by the SPOR domain are shown, with the residues involved in PG recognition labeled and those contributing only through main-chain interactions given a number. Relevant residues and crystallographic water molecules are shown as capped sticks and spheres, respectively. Polar interactions are represented as dotted lines.

concern side chains of residues R302 and R309, which offer two distinct conformations with direct involvement in ligand recognition and F274 that slightly changes its orientation upon ligand binding (Supplementary Fig. 7). This limited motion is supported by our dynamics simulations (vide infra). (ii) Notwithstanding that all four saccharides interact with the protein, interactions are not evenly distributed among the four subsites. The largest number of interactions is observed for NAM(1) (eight polar interactions with four atoms in the NAM), and to a lesser extent with NAG(2) (four polar interactions with three atoms in the NAG) and NAM(3) (five polar interactions with four atoms in the NAM). A single polar interaction is seen for NAG(4) (Fig. 4c). (iii) The carboxylate moieties of the NAM rings are strongly recognized by the SPOR domain, as described in the section on dynamics simulations later. (iv) Several amino acids participate in the stabilization of the tetrasaccharide, but only the side chains of Q270, R302 and R309 are directly involved in the interaction with the glycan chain through polar interactions with oxygen atoms of the carboxylate of the lactyl moiety and the N-acetyls of NAM(1) and NAG(2). It is noteworthy that these residues, together with R311, build a basic patch critical for the binding of the carboxylate and acetyl groups of NAM(1). (v) Main-chain interactions are essential in PG stabilization by SPOR domains. Most of residues contribute to the stabilization of the glycan chain by the formation of hydrogen bonds with the protein backbone (residues 271, 273, 275, 276, 333, and 335) or by stabilization of the hydrogen-bond network mediated by water molecules (Fig. 4c and Supplementary Fig. 6). (vi) hydrophobic interactions are observed with F274 and L307. As explained below, these results also shed light on previous studies that tackled the interaction between

SPOR proteins and PG using localization assays by fluorescence microscopy and PG binding assays by NMR[5,19,20,22].

The crystal structure of SPOR-RlpA:**3** complex was solved at 1.4 Å resolution (see Supplementary Table 2 and Material and Methods). Compound **3** occupies the four saccharide-binding subsites with anhNAM at position 1 (Fig. 5a). While the remaining saccharides (positions 2–4) mirror the pattern of interactions observed for compound **1**, the binding of the anhNAM to the SPOR domain is different. In this case, five polar interactions are observed with two atoms (vs. eight interactions with four atoms in NAM(1)) between the lactyl carboxylate and R302, R309, and Q270. Interestingly, new water molecules are seen in this complex linking the lactyl carboxylate with the new position of the acetyl group (Fig. 5b). The arginines within the basic pocket change their conformation upon binding of **3** to account for the structural change of NAM to anhNAM in the PG. A decrease in the number of interactions at position 1 allows for mobility of the anhNAM ring, as reflected by the increase in the B factors (Supplementary Fig. 8 and Supplementary Fig. 2C). These results suggest that SPOR domains are capable of binding peripheral PG (containing the anhNAM) by retaining the interactions with the saccharide at the other subsites and by changing the conformation of the arginine side chains at the basic patch to promote interactions with the carboxylate group of the anhNAM. Our mass-spectrometry assays demonstrated that PG derivative **1** with NAM(1) in its structure exhibits better binding than **3** with anhNAM(1) to SPOR-RlpA, which is consistent with our structural observations described here.

Structural comparison of PG complexes in two pseudomonal LTs—the SPOR domain of RlpA and the Slt[32]—provides insights on specific recognition of PG by SPOR domains. While Slt

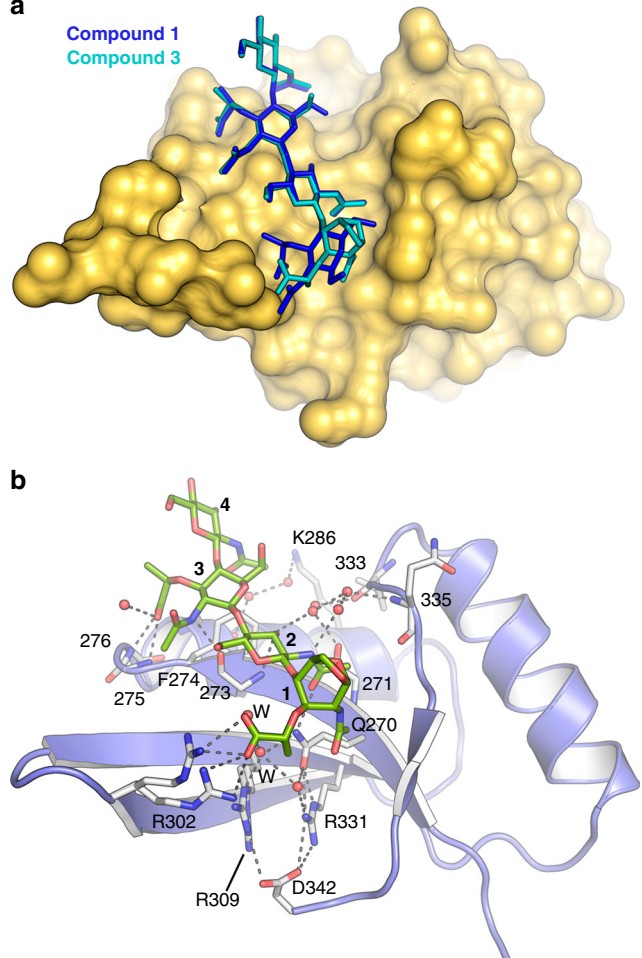

**Fig. 5** Crystal structure of the SPOR-RlpA in complex with anhNAM (**3**). **a** A surface representation of the SPOR domain (colored in yellow) in complex with compound **3** (green sticks) is shown. The structure of compound **1** (blue capped sticks) as observed in the SPOR-RlpA:**1** complex is superimposed (protein omitted for clarity). **b** The recognition of the anhNAM-modified PG by SPOR-RlpA domain is depicted. The residues involved in PG recognition are labeled and the amino-acid numbers are given for those contributing through main-chain interactions. The relevant residues and water molecules for the interaction are presented as capped sticks and spheres, respectively. Polar interactions are represented as dotted lines.

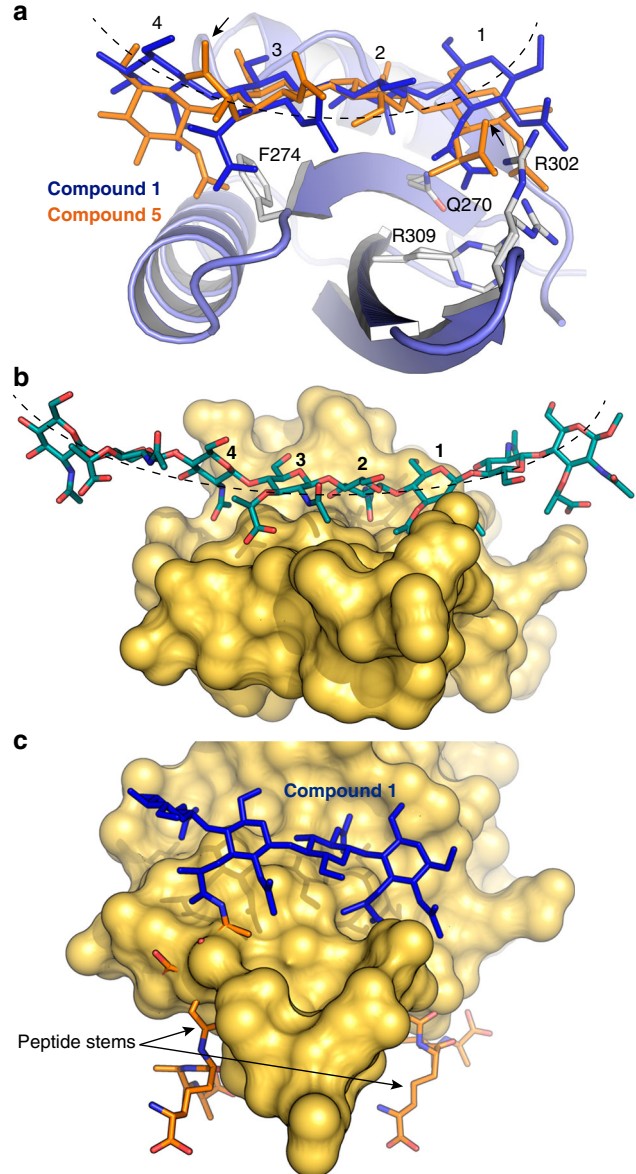

**Fig. 6** Peptide stems abrogates SPOR-RlpA binding to PG. **a** Structural comparison of the SPOR-RlpA:**1** complex (blue ribbon and blue capped sticks) with that of compound **5** (orange capped sticks) observed in the Slt:**5** complex (PDB code 6FCS). The side chains of the amino acids defining the peptidoglycan-binding site are represented as capped sticks with gray carbon atoms. The peptide stems in **5** have been omitted for clarity, with their points of attachment to the NAM units in **5** indicated by arrows. **b** The extended model for glycan chain interaction with SPOR-RlpA domain. **c** A hand model for the SPOR-RlpA:**1** complex with the peptide stems attached to the lactyl groups. The peptide conformation is according to PDB code 6FCS. The ligand **1** is represented as blue capped-sticks and the modeled peptide stems are in orange for the carbon atoms; the peptide stems run through the protein in this model. The dashed lines on (**a** and **b**) were manually generated to highlight the curve of the glycan chain.

presents a long groove embracing the substrate with many residues interacting through their side chains with both glycan and peptide moieties[32], SPOR domain presents a short and open binding site that recognizes glycan chain mainly through main-chain interactions. The PG-binding site is limited by F274 residue on one side and by the cluster Q270, R302 and R309 on the other (Fig. 6a). Superimposition of compound **1** (this work) and compound **5** as observed in the Slt complex (PDB code 6FCS) (Fig. 6a) indicates that PG conformations are different. The extremes of SPOR binding site (F274 and basic cluster) provoke a bent conformation of glycan chain at positions 1 and 4 compared with the glycan chain in the Slt:**5** complex (Fig. 6a). This curvature is seen best if we extend the glycan chain by two additional glycans at the two termini (by superimposing the same structure for the structure of **1** at positions 1, 2 or 3, 4) (Fig. 6b). Interestingly, this bent conformation for the denuded glycan chain abrogates the opportunity for any further interactions

between the SPOR domain and the PG beyond the four sugars recognized at the center of the domain.

The structural basis for why the SPOR domain does not recognize PG chains with peptide-stems can now be explained. In agreement with our mass-spectrometric experiments that clearly noted the absence of binding of compound **5** to the SPOR domain (Fig. 3), the crystal structure of SPOR-RlpA:**1** reveals that

recognition of the carboxylate moieties of muramic acids at positions 1 and 3 is crucial (Fig. 4c and Supplementary Fig. 6). These carboxylate moieties become available only after the removal of peptide-stems by the action of the aforementioned amidases. If peptide-stems were included in the SPOR-RlpA:**1** complex (Fig. 6c), the strong steric clashes with the SPOR structure would preclude binding by the PG; which is indeed the observed outcome.

To further validate our binding model, we perform site-directed mutagenesis. Q270 and A273 have been already shown to be important in SPOR-DamX[20] and SPOR-FtsN[19], thus we concentrated on the previously undescribed basic patch (R302, R309, and R311). Each arginine was converted to alanine individually (R302A, R309A, and R311A). In addition, we generated the triple variant, in which the three arginines we mutagenized to alanine (see Materials and Methods). We also performed mutagenesis for residues Q270 and F274 by converting them to alanine individually. Unfortunately, the corresponding proteins could not be produced for analysis by mass spectrometry as they required addition of detergents to increase solubility, so they were not studied further. Compound **1** was used to test the binding avidity to the four SPOR-RlpA domain variants of the arginines in the cluster. The three single-amino-acid variants of SPOR-RlpA domain (R302A, R309A, and R311A) showed decreased binding of compound **1** (Fig. 7d, f, and h), compared to the wild-type SPOR-RlpA domain (Fig. 7b). A binding event was observed at a molecular mass of 12,021 Da for the R302A and R309A variants. The mass shift (+16 Da) observed in the case of the R311A variant is due to air oxidation of a methionine. In case of the triple variant (R302A/R309A/R311A), the binding was decreased significantly (Fig. 7j). These results argue that the cluster of triple arginines serves a cushioning function, as the site is important for recognition of the PG. A single arginine alteration by mutation of the gene would not abrogate recognition as the other two compensate. But the loss of all three is very detrimental.

**Interactions of tetrasaccharide 1 and modified PGs with SPOR-RlpA.** In order to investigate the dynamics behavior of the SPOR domain in complex with PG fragments, we conducted molecular-dynamics (MD) simulations with AMBER16[33] program. We first simulated the SPOR-RlpA:**1** complex for 500 ns by a previously described method[34] (see Methods for details). Despite the shallowness of the PG-binding groove, the simulation demonstrated stable association of **1** with the SPOR domain throughout the simulation (Fig. 8 and Supplementary Movie 1). The carboxylate of the lactyl moiety of NAM(1) inserts into the pocket formed by the side chains of the three arginines that provide an electrostatic interface. Among the three arginine residues, R309 maintained the most stable interaction to **1** with direct hydrogen-bonding, while for R302 we see considerable fluctuation (Fig. 8). Residue R311—positioned deeper within the pocket—indirectly interacted with **1** via hydrogen bonding with water molecules. The highly conserved Q270 provided hydrogen bonds through its side chain amide nitrogen to the lactyl group of NAM(1) and the N-acetyl carbonyl oxygen of NAG(2) (named Q270-COO and Q270-N, respectively, in Fig. 8). Notably, the Q270-N hydrogen bond consistently persisted over the length of simulation, suggesting a key role in binding to PG. In addition, hydrogen-bonding interaction with the backbone of A273, F274, A275, and N276 stabilized NAG(2) and NAM(3) of the tetrasaccharide. Another relevant contact was provided by F274, which formed favorable CH/π interactions with NAM(3)—widely present in carbohydrate recognition sites of proteins[35,36]. The most mobile region of **1** was that of the NAG(4), which by itself did not

establish any significant direct contact with the protein at any point of the simulation.

PG is known to undergo several modifications to its saccharide segments, most of these modifications being organism-dependent (Fig. 2b). Three modifications are known in Gram-negative bacteria and entail the conversion of NAM to 1,6-anhNAM (product of the lytic transglycosylases as in compound **3**), the formation of C6-O-acetylated NAM (potentially a regulatory mechanism to halt the reactions of lytic transglycosylases), and N-deacetylation (typically of NAG) (Fig. 2b). N–Deacetylated and/or O–acetylated glycan strands in pathogenic species act as a defense mechanism against lysozymes from the host[37], the latter is also proposed to be important for proper cell growth and fitness[38]. We performed dynamics simulations of the SPOR domain in complex with each of these modified PG structures, based on the collective knowledge from our X-ray structure complexes. The dynamics simulations of SPOR-RlpA:**3** complex (X-ray) revealed a pattern similar to that seen in the simulations of the SPOR-RlpA:**1** (X-ray) complex, mentioned above (Supplementary movie 2). The lactate carboxylate of 1,6-anhNAM(1) interacted with the arginine side chains, while the NAG(2) maintained hydrogen bonds with the backbone amide of A273 and side-chain of Q270 (Supplementary Fig. 9, A273-N and Q270-NAc). Interestingly, the rigid 1,6-anhNAM(1) appeared to affect the dynamics of the NAM(3) and NAG(4). The NAG(4) demonstrated a remarkable increase in mobility, compared to that of the simulation for SPOR-RlpA:**1**. This is in large measure due to the inability of NAM(3) to establish a hydrogen bond with the backbone of A275 and N276 (Supplementary Fig. 9). The structural differences between anhNAM(1) in compound **3** and NAM(1) in compound **1** are of note. The ring substituents in NAM are all equatorial, whereas in anhNAM they are all axial. This configurational difference places the lactate carboxylate in distinct spatial locations in the two complexes. Yet, the carboxylate in both cases is matched with the electrostatic patch created by the side chains of the three arginines. The dynamical mobility of the side chains of these arginines accommodates the distinct spatial disposition of the lactate carboxylate in the two PG-derivatives and is likely an evolutionary adaptation for the SPOR domain to be able to bind to both kinds of PG.

Next, we computationally modeled two independent complexes of SPOR-RlpA, corresponding to O-acetylated and N-deacetylated PG modifications. The SPOR-RlpA:**1** co-crystal structure was used as template for modeling each of these complexes. The C6-O-acetylated PG (acetyl group modification on NAM(1) and NAM(3)), and N-deacetylated PG (N-deacetylation at NAG(2) and NAG(4)) were modeled to SPOR-RlpA:**1** and energy-minimized. The models were subjected to MD simulation. The C6-O-acetyl functionality was exposed to the solvent and it was highly flexible during most of the simulation. It did not appear to have a notable effect on the binding of the O-acetylated-PG ligand (Supplementary movie 3). In contrast, the complex of SPOR-RlpA with N-deacetylated PG promptly dissociated. Considering that physiological $pK_a$ for glucosamine is 8.12[39], neutral-amine N-deacetyl-PG ligand was also considered in our simulations. In both cases (charged or neutral amine) the N-deacetyl-PG did not remain in the binding pocket. Interestingly, the N-deacetylated PG was not able to form the hydrogen bond with Q270, compared to the other PG modifications that we investigated. In the other complexes, hydrogen bond of Q270 with the N-acetyl group of NAG(2) was maintained consistently during the simulation (Q270-NAc, Fig. 8 and Supplementary Fig. 9). Thus, our simulations suggest that binding of deacetylated PG to SPOR domain is unfavorable and

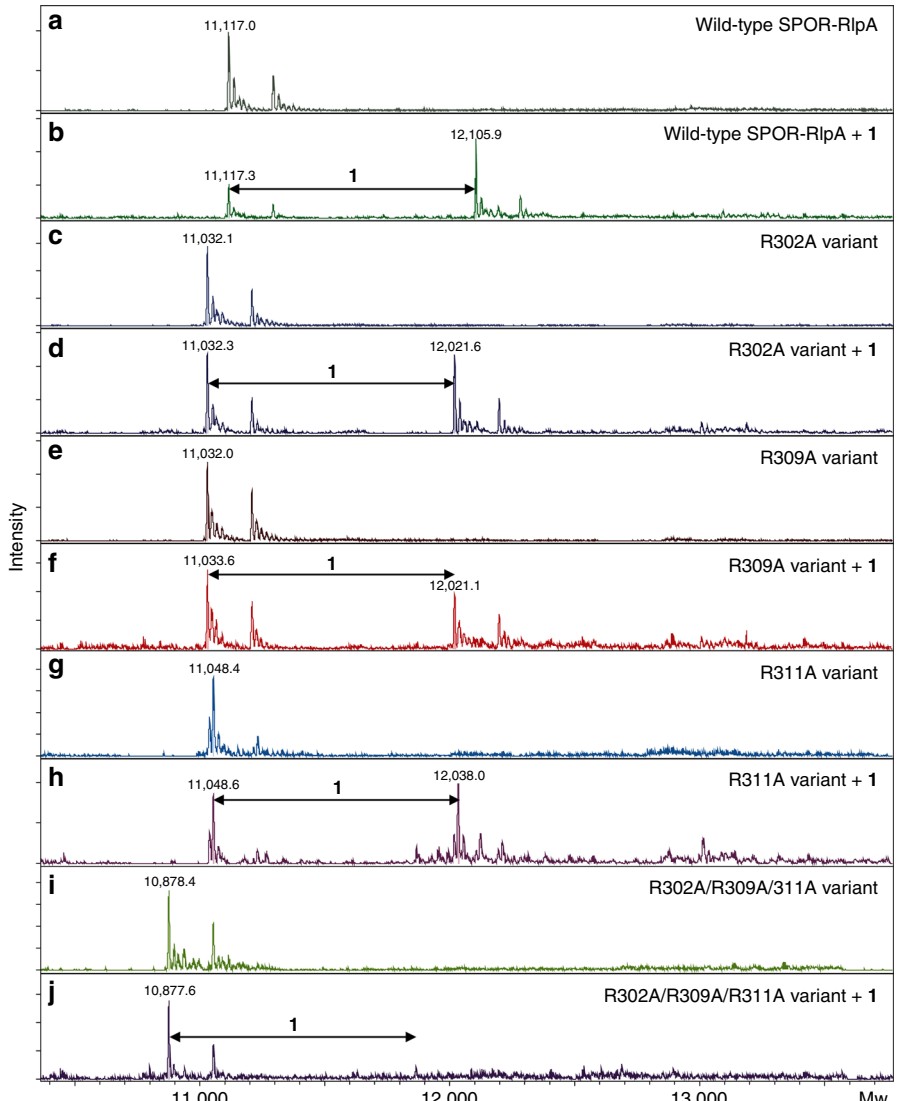

**Fig. 7** Binding of SPOR-RlpA variants to compound **1**. Representative deconvoluted electrospray mass spectra displaying the SPOR-RlpA domain wild-type and four variants as an intact monomer (**a**, **c**, **e**, **g**, and **i**) and in complex with synthetic muropeptide **1** (**b**, **d**, **f**, **h**, and **j**). **a**, **b** Wild-type SPOR-RlpA domain, **c**, **d** R302A, **e**, **f** R309A, **g**, **h** R311A, and **i**, **j** R302A/R309A/311A triple variant of the SPOR-RlpA domain. The intensity of the signal is plotted *vs* Mw (average molecular weight) of the indicated species.

therefore, according to the simulation, *N*-deacetylated PG is not likely to be recognized by SPOR-RlpA.

## Discussion

As stated earlier, NMR structures have been solved for three distinct SPOR domains, however these structures do not shed light on how PG is recognized. Our crystal structure of SPOR-RlpA:**1** complex provides both the molecular basis for such an understanding and the structural support of previous functional studies by site-directed mutagenesis, followed by septal localization of the GFP fusion protein and/or by NMR chemical-shift studies in the presence of bacterial sacculus[20–22].

The structural comparison of our complexes with the NMR SPOR domains, together with analysis of the functional studies in the light of our results, allow identification of key conserved features in denuded glycan recognition by SPOR domains (Fig. 9). The side chains of four amino acids in SPOR-RlpA— those of R302, R309, R311, and Q270—establish direct polar contacts with the PG. The cluster of the three arginines

generates a basic patch along the β1-β2 region involved in interactions with NAM(1) (or anhNAM(1)). This basic patch is strictly preserved in the other SPOR domains, although the residues that create the surface are not (R380 and K418 for DamX, R285 for FtsN, K47 and K42 for CwlC) (Supplementary Fig. 10 and Supplementary Table 3). Only Q270, responsible for interactions with NAM(1) and NAG(2), is highly conserved among the sequence of SPOR domains of known structures (Supplementary Fig. 11A and Supplementary Table 3). Sequence alignment comprising all of the available sequences coding for SPOR domains by a profile-hidden Markov model (profile HMM) shows that a glutamine occupies preferentially this position (position 8 in Supplementary Fig. 11B). Another interesting observation in HMM profile is that position 12 (Supplementary Fig. 11B) is preferentially occupied by a phenylalanine residue (F274 in SPOR-RlpA) that aligns with NAG (4) in SPOR-RlpA:**1** complex, which imposes a curvature onto the glycan chain. While few side chains are involved in PG stabilization in SPOR-RlpA, most of the polar contacts stabilizing the four glycan rings are by the protein backbone. These

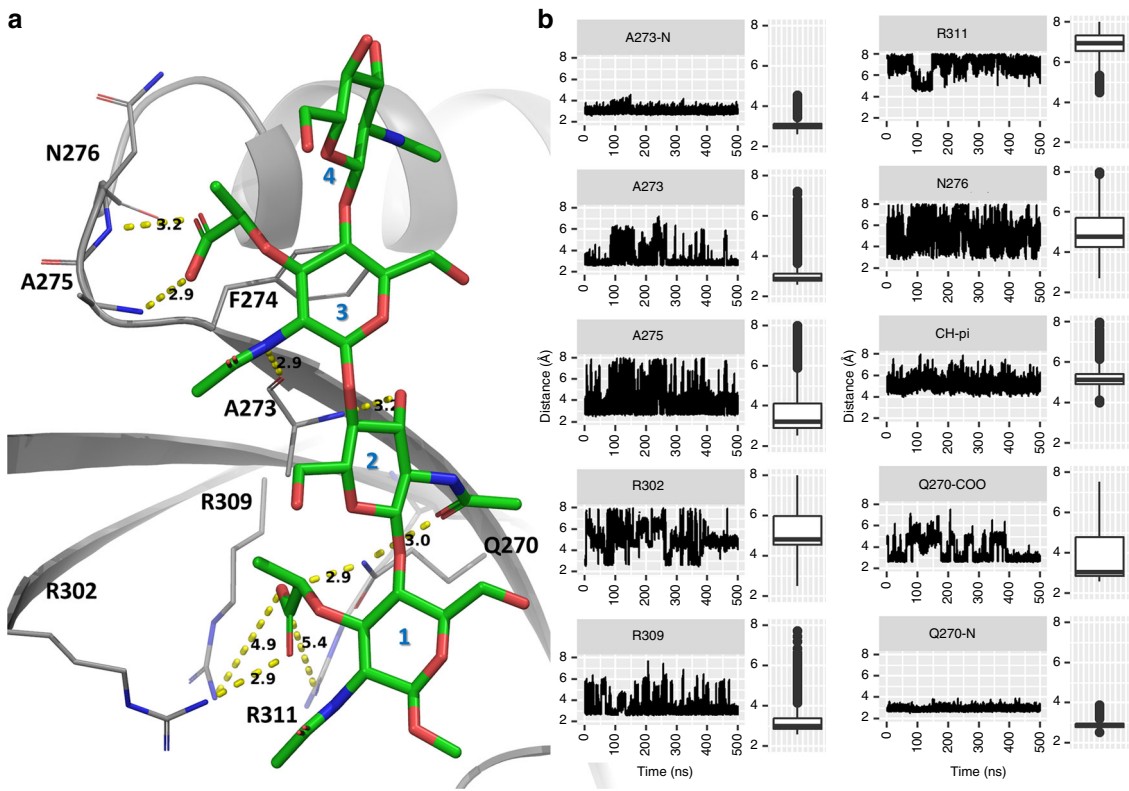

**Fig. 8** Dynamic interactions that stabilize denuded PG (**1**) with SPOR-RlpA, as calculated by molecular-dynamics simulation. **a** The interactions of **1** with SPOR-RlpA are depicted per the energy-minimized X-ray complex. **b** The dynamics fluctuations in intermolecular interactions are shown as distances between protein and ligand atoms in the line-plots and box-plots (on the right) calculated for ten thousand sampled conformations of the 500 ns MD simulation. The boxes in the middle of the box plots show the middle 50% of the sampled distances, while the upper and lower lines of the box represent upper and lower 25% of the distances. The horizontal lines inside the box denote the median value. The dots at the end of the lines, if present, are outliers. For consistency in plotting the y-axis, the values in the range of 2–8 Å of the distances are plotted in all the figures. N276, A275: Distance between carboxylate oxygen* of NAM(3) and main-chain nitrogen atoms of residues N276 and A276 respectively. A273: Distance between 3-hydroxyl oxygen atom of NAG(2) and main-chain nitrogen atom of A273.A273-N: distance between main-chain oxygen atom of A273 with acetamide nitrogen atom of NAM(3). R302, R309, and R311: Distance between the oxygen* atom of the carboxylate group of NAM(1) with the nitrogen** atom of arginine residue. (Indicated by yellow broken line in the Fig. 6a).CH-pi: Distance between the centroid of the aromatic ring of F274 and centroid of three of the carbon atoms (C1, C3, and C5) of NAG(3). *Any of the two oxygen atoms of the carboxylate group with least distance to interacting nitrogen atom. **Any of the three nitrogen atoms of each arginine residues with least distance to interacting oxygen atom.

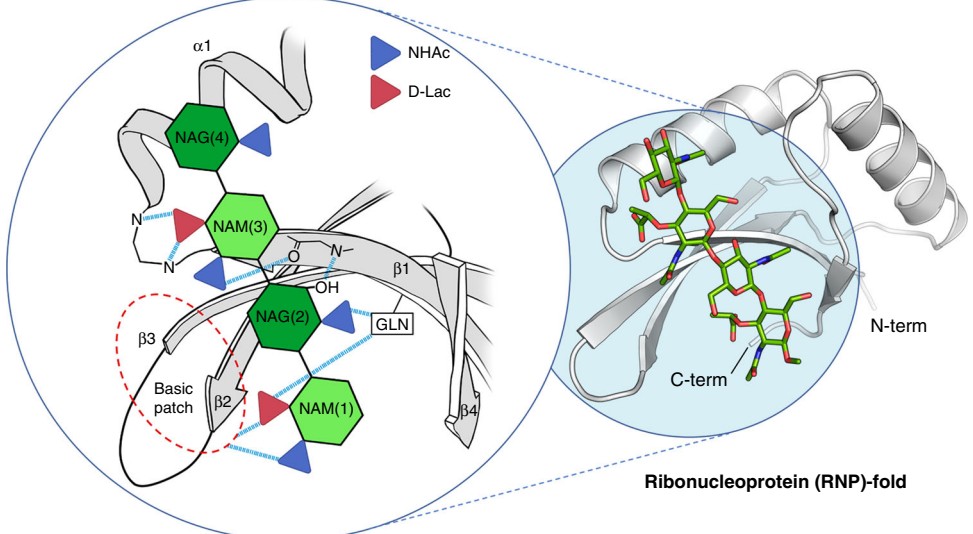

**Fig. 9** Common principles in PG recognition by SPOR domains. A schematic model for denuded-glycan recognition by SPOR domains is depicted. NHAc and D-Lac stand for *N*-acetyl and lactyl groups, respectively. Polar interactions are represented by dashed lines.

results explain why the sequence conservation is not seen among SPOR domains, but the conservation of the three-dimensional architecture of the protein is. For example, the backbone nitrogens of A275 and N276 (connecting β1 with α1), create a structural feature conserved among all the available SPOR structures, allowing the formation of two polar contacts with the carboxylate of the D-lactyl moiety of the NAM(3) (Fig. 8). Both the main-chain nitrogen and carbonyl oxygen of A273 from β1 and the carbonyl oxygen of V271 (RlpA numbering) are also involved in polar contacts with NAG(2) and NAM(3). These interactions depend on main-chain atoms and thus do not depend on specific residues at these positions. The β-strand, which is conserved in SPOR domains, will likely lead to conservation of interaction with the glycan across SPOR domains. In agreement with this view, two previous functional studies implicate the β-strand in PG binding[19,20]. Duncan et al.[19] performed extensive mutagenesis of the 33 residues that are distributed throughout the SPOR-FtsN surface, concluding that only five amino acids where shown to reduce septal localization when mutagenized (all located in the β-sheet, where PG binds): Q251, S254, W283, R285, and I313. The physiological relevance of these residues can be nicely explained with our structural model. Q251 is homologous to Q270 in SPOR-RlpA, therefore presumably interacts with one the lactyl carboxylate of NAM (1). R285 is homologous to SPOR-RlpA R309 and would be involved in the electrostatic interaction with the NAM (1). S254 has no homolog in SPOR-RlpA (instead A273), but according to its location, it could be involved in the formation of a hydrogen bond with the N–acetyl group of NAG (2). In addition, it could establish the same polar contacts involving atoms from the backbone with the glycan chain. This is supported by the fact that the S254A mutant does not affect septal localization, pointing that the side chain is not relevant[19]. W283 is also not present in SPOR-RlpA (L307 is found instead). This tryptophan could play a role in stabilization of methyl groups in NAM(1) and NAM(3) through hydrophobic interactions, as also happens in SPOR-RlpA by L307 (Fig. 4c, Supplementary Fig. 6D and Supplementary Table 3). The I313 is not conserved in SPOR-RlpA, but according to our model, would be close to the methyl group of NAG(2) for hydrophobic interactions.

It is of note that in SPOR-FtsN the binding site is partially occluded by the loop connecting α2 with β4 (residues 308–313) (Supplementary Fig. 10B). Interestingly, SPOR-FtsN presents two Cys residues (C312 and C252) in this region that could rearrange upon disulfide bond formation, nicely explaining experimental correlation between PG binding and disulfide bond formation in FtsN[19].

In SPOR-DamX, Williams et al.[20] showed that three residues play a key role in septal localization: Q351, S354 and W416. Q351 is homologous to Q270 in SPOR-RlpA, so very likely plays the same role. S354 occupies the equivalent position as A273 in SPOR-RlpA (and S254 in SPOR-FtsN). Interestingly, Williams et al.[20] showed that the side-chain hydroxyl is not required for septal localization, reinforcing the idea that the main role of the residue located at this position is the interaction with the glycan strand through backbone atoms, as observed in SPOR-RlpA. W416 is located in the β-sheet, but is not homologous to W283 of SPOR-FtsN (nor L307 in SPOR-RlpA), which is in β3, whereas W416 of SPOR-DamX is in β4. Superimposition of SPOR-DamX structure onto our SPOR-RlpA:1 complex (Supplementary Fig. 10), reveals that W416 is optimally placed for stabilizing the NAM (1) by aromatic stacking. Interestingly, our model points to other additional residues among SPOR domains from different origins that could play a role in reinforcing glycan interactions (see Supplementary Fig. 10 and Supplementary Table 3). For instance, W364 in DamX could play a role in the interaction with

the glycan chain, as also observed in other lytic transglycosylases[40] and endolysins[41]. A summary of the residues involved in glycan recognition for the SPOR domains in CwlC, FtsN and DamX, as predicted from the SPOR-RlpA complexes, is described in Supplementary Table 3.

All SPOR domains whose structures have been determined to date present a similar fold and, overall, exhibit less than 20% amino-acid sequence identity. Three-dimensional structures of the complexes reported here, together with molecular dynamics simulations and mass spectrometry experiments with authentic PG samples, provide the first clues about how the large family of SPOR domains, distributed over 4513 species of both Gram-negative and Gram-positive bacteria, can share a common pattern for recognition of the septal PG.

Previous structure-function studies for SPOR-DamX[20] and SPOR-FtsN[19] showed that glycans binds at the β-sheet of SPOR, as revealed by mutagenesis studies. Our structural study reveals a wealth of new detail about the SPOR-glycan interaction that mere genetic studies could not. One of the more interesting findings in the present report is that many of the protein-ligand interactions involve main-chain atoms rather than side-chain functions. This nicely explains how multiple SPOR domains with very little sequence conservation can nevertheless bind to the same glycan ligand. The structures also reveal for the first time the chemical features of the glycan that are important for binding. Thus, we can now understand at atomic detail how only denuded glycans bind to the domain. It turns out that the if the peptide stem were present, it would sterically block binding to the SPOR domain. Enzymatic removal of the stem peptide exposes the charged lactyl carboxylate of NAM, which interacts with the conserved glutamine identified in previous studies, and the positively charged patch of the three arginines.

SPOR domains of various species conform to the requirements for recognition of denuded PG by a combination of hydrogen-bond networks to main-chain atoms (as a consequence of the specific RNP fold) mediated by water molecules or by polar and salt-bridge interactions provided by very few critical residues (Q270 and arginines at the basic patch). This specificity for denuded PG is strict. However, as reported herein, some PG modifications on denuded glycans are also allowed for recognition by SPOR domains. Binding of the end of the PG strands through the 1,6-anhNAM, as a consequence of the action of lytic transglycosylases, or O-acetylation on NAM units is also allowed while N-deacetylation on NAG units are not.

During cell division there is a concerted coordination of distinct processes involved in PG biosynthesis, inner- and outer-membrane constriction, and synchrony in PG degradation in a manner that lends itself to the formation of daughter cells (Fig. 1b). This orchestration suggests that all these events take place in close proximity[30]. Proteins harboring the SPOR domain are located at the septal ring under a precise spatiotemporal regimen for recruitment of the individual protein. This regulation is, at the minimum, dependent on the activity of PG amidases that cleave peptide stems in both crosslinked and non-crosslinked PG chains at the septum (Fig. 1c). This model was first proposed by Gerding et al.[13] Noteworthy extensions of the concept were reported by Busiek and Margolin[42], who showed that FtsN initially localizes to division sites in a SPOR-independent manner that then precedes with its SPOR-dependent localization. Yahashiri et al.[5] provided experimental support for the role of LTs in releasing SPOR proteins from septal PG. In addition, a recent review by the David S. Weiss group provided a more nuanced treatment of the model[14]. The coordination of the envelope machinery for PG biosynthesis and outer-membrane constriction[30] and the control of the activity of PG amidases through divisome-associated activators[43] together

with the strict recognition mechanism by SPOR-containing proteins provide an exquisite spatiotemporal control of critical proteins during cell division.

It is of note that despite relevance of lytic transglycosylases in different aspects of bacterial fitness[25] RlpA of *P. aeruginosa* and LtgC of *Neisseria gonorrhoeae*[44] are the only examples known so far whose deletion presents a specific phenotype—formation of filamentous chained cells—as a result of an inefficient separation of daughter cells[23]. A covalently attached lipid at the amino terminus of RlpA allows the anchoring of the protein in the inner leaflet of the outer membrane, whereas its SPOR domain orchestrates the substrate recognition by the catalytic domain; that are, the denuded glycans that is sequestered at the cell-division septum[5,23] (Fig. 1b, c). We speculate that this arrangement might reflect a conformation of RlpA in which its SPOR domain would properly present the denuded glycan to the catalytic DPBB. The two extremes of the binding site (the F274 positions and the basic cluster) act as wedges that limit binding to merely four glycan rings in the substrate PG. According to our structural model, recognition by SPOR domains requires that at least two consecutive NAM-NAG units to be devoid of their peptide stems. Therefore an extensive footprint for the activity of PG amidases would not be required for the generation of the PG stretch specifically recognized by SPOR domains. This notion is in agreement with the fact that denuded glycans are a moving target due to their transient nature. They link critically the sequestration of a SPOR protein, and thus its activity by entropic reasons, to the site of cell division.

Yahashiri et al.[5,14] demonstrated localization of SPOR domains on purified *sacculus*, which by necessity excludes the possibility that the SPOR domains targeting a distinct septal protein, rather than a distinct PG. However, the work that we disclose herein provides the direct evidence, by using a homogeneous and chemically defined PG ligand, that SPOR-RlpA binds denuded PG to the exclusion of PG with peptide stems. Our work also provides the structural basis for the requirement of a denuded glycan substrate.

As the proteins with the SPOR domain are located at the septal ring, our experiments enforce the previously stated[5] notion that denuded PG are enriched in that locale, which was an inference, but not an establish fact due to the difficulty of isolating septal PG. We addressed this paucity of information in the present study by the synthetic PGs.

Considering that bacterial mutants lacking SPOR domain proteins are often sensitized to several antibiotics or are conditionally lethal[12,23,45–48], our structures of SPOR-RlpA complexes may pave the way for development of small-molecule inhibitors with therapeutic potential. Future three-dimensional structure determination of other SPOR domains in complex with PG and possible regulatory partners will provide further valuable information for the elucidation of the molecular mechanisms underlying regulation in bacterial division.

## Methods

**Synthesis of PG derivatives**. PG derivatives were prepared according to the literature methods developed by our laboratory[8,49–51].

**Non-denaturing mass spectrometry**. The methodology used in this work for the non-denaturing mass spectrometry has been described previously[52,53]. Briefly, the wild type or any variant SPOR-RlpA (20 µM) was buffer exchanged into 1 M ammonium acetate, pH 6.6, by the use of a Zeba Desalting Column (Thermo Fisher Scientific). To remove the remaining salts from the protein, the sample was subsequently buffer exchanged into 1 M ammonium acetate, pH 6.6, (1:200 dilution) using an Amicon Ultra 0.5 mL Centrifugal Filter (3 kDa molecular weight cut-off). Electrospray mass spectra of the SPOR-RlpA (20 µM) and synthetic muropeptide (500 µM) complexes (obtained after 30 min incubation time) were acquired using a MicrOTOF-QII (Bruker, Billerica, MA) under conditions optimized for observation of non-covalent protein complex interactions in which a ligand excess

concentration is used to allow the observation of weak interactions and also saturate the binding site. Typical instrument parameters, in positive-ion mode, on the MicrOTOF-QII for the SPOR-RlpA (wild type or variant) were: end plate offset −0.5 kV, capillary voltage 4.5 kV, nebulizer gas pressure 0.4 bar, dry gas flow rate 4.0 L/min, dry gas temperature 180 °C, funnel 1 RF 300 V, funnel 2 RF 400 V, hexapole RF 400 V, quadrupole ion energy 5 V, collision energy 8 eV, collision cell RF 1.2 kV, ion transfer time 130 µs, and pre-pulse ion storage 25 µs. Solutions were infused at a flow rate of 3 µL/min.

**Cloning and purification of SPOR-RlpA domain**. The cloning and purification of the full-length RlpA from *P. aeruginosa* PAO1 was previously reported[27]. The plasmid pET-28a(+) (Novagen) containing the *rlpA* wild-type gene was used as the template for the PCR reaction to produce the SPOR-domain clone (amino acids 264–342). We used Q5 High-Fidelity DNA Polymerase (NEB) with primers listed in Supplementary Table 1 and restriction enzymes NdeI and XhoI. The gene was cloned into a pTEV15b vector (gift from the J. M. Pereda lab, IBMCC, Salamanca, Spain), which is a derivative of the pET-15b vector with two modifications: (i) the 6xHis-tag has been increased to 8xHis-tag and (ii) the original thrombin site (LVPRGS) has been changed to a TEV protease cleavage site (ENLYFQG), which allows the removal of the N-terminal tag (MGSSHHHHHHHHSSGENLYFQ). The plasmid pTEV15b_SPOR-RlpA wild-type was used as template for site-directed mutagenesis to produce all mutants. The triple mutant R302A/R309A/R311A was prepared in two subsequent steps: (i) using pTEV15b_SPOR-RlpA-R311A as template with primers 2 (Supplementary Table 1); (ii) the double mutant as the template for the third couple of primers 5 (Supplementary Table 1). The ligation was performed using the KLD kit from New England Biolabs. The purity of the PCR reaction products was assessed by agarose-gel electrophoresis and the cloning results were confirmed by DNA sequencing after each step on both strands. Competent *Escherichia coli* BL21 Star (DE3) (Invitrogen) were transformed with the pTEV15b_SPOR-RlpA plasmid. The overexpression and purification of SPOR-RlpA were performed as previously described for the full-length RlpA protein[27] with slight modifications. Briefly, 5 mL of Ni-NTA resin (Macherey-Nagel) was used for the purification and the bound protein was eluted using from the resin in an imidazole gradient up to 500 mM. After purification and removal of the His-tag (using TEV protease), the protein was concentrated (using an Amicon Ultra 15 mL Centrifugal Filter; 3 kDa molecular weight cut-off) to a final concentration of 8 mg/mL (see gel on Supplementary Fig. 1A for SPOR-RlpA wild type as an example of protein purity). The final yield of the purification was approximately 10 mg of protein per 1 L of culture. The genes for two mutants, Q270A and F274A, were successfully cloned and expressed. However, the solubility of the resultant protein was dramatically altered, requiring the presence of detergents. This precluded the ability to test binding of the proteins with the PG ligands by mass spectrometry.

**SPOR-RlpA crystallization**. Crystallization screenings were performed by high-throughput techniques in a NanoDrop robot and Innovadyne SD-2 microplates (Innovadyne Technologies Inc.), screening PACT Suite and JCSG Suite (Qiagen), JBScreen Classic 1–4 and 6 (Jena Bioscience) and Crystal Screen, Crystal Screen 2 and Index HT (Hampton Research). The conditions that produced crystals were optimized by sitting-drop vapor-diffusion method at 291 K by mixing 1 µL of protein solution and 1 µL of precipitant solution, equilibrated against 150 µL of precipitant solution in the reservoir chamber. The best crystals were obtained in a crystallization condition containing 0.15 M NaF and 16% (w/v) PEG3350 (Supplementary Fig. 1B). Protein concentration was assayed at the concentration range of 7–8 mg/mL.

**Soaking and co-crystallization experiments**. For both soaking and co-crystallization trials, compounds **1**, **2**, and **3** were dissolved in water and incubated with native protein crystals at a final concentration of 5 mM using the crystallization conditions described above. Soaking experiments were incubated overnight at 291 K.

**Data collection and structural determination**. Crystals were cryo-protected in the precipitant solution supplemented with 30% (v/v) glycerol, prior to flash cooling at 100 K. Diffraction data was collected in beamline XALOC at the ALBA synchrotron (CELLS-ALBA, Spain), using a Pilatus 6 M detector and a wavelength of 0.979257 Å (Supplementary Fig. 1). Crystals diffracted up to 1.2–1.6 Å resolution and belonged to the C 2 2 2$_1$ space group, being the unit cell parameters $a = 67.58$ Å, $b = 68.76$ Å, $c = 38.77$ Å, $\alpha = \beta = \gamma = 90°$. The collected datasets were processed with XDS[54] and Aimless[55]. One molecule of SPOR-RlpA was found in the asymmetric unit, yielding a Matthews coefficient of 2.70 Å³/Da[56] and a solvent content of 54.43%.

Structure determination was performed by de novo phasing with Arcimboldo[31]. We used a search including two copies of a helix containing 13 residues assuming RMSD from target of 0.2 Å. Refinement and manual model building of SPOR-RlpA was performed with Phenix[57] and Coot[58], respectively. The stereochemistry of the final model was checked by MolProbity[59]. The ligand-bound structures were determined by molecular replacement of the apo SPOR-RlpA form. For the structures with bound glycan ligand, CIF files for each ligand were generated using eLBOW[60]. In all cases refinement strategy included atomic coordinates, individual

B-factors, TLS parameters, occupancies and automatically correction of N/Q/H errors. After refinement, all glycan structures where validated with the program Pivateer[61], using a mask radius around the sugar atoms of 1.5 Å. Data collection and processing statistics are shown in Supplementary Table 2.

**Computational modeling and simulation method.** The crystal structures of the RlpA SPOR domain (SPOR-RlpA:**1** and SPOR-RlpA:**3**) were prepared for simulations using the protein preparation wizard module of the Schrodinger Suite (v 2015, Schrodinger Inc., NY). Hydrogen atoms were added and bond orders were assigned. The charges for the atoms of all the ligands were calculated quantum-mechanically at HF/6–31 G(d) level of theory (Gaussian program v. g09, Gaussian, Inc., Wallingford CT, 2016) and were fit with RESP methodology[62] (antechamber program, AMBER16[33]). The complexes were immersed in water molecules (TIP3P model) of a truncated octahedron box with dimensions such that the protein atoms were at least 15 Å away from any of the box edges. The system was energy-minimized, equilibrated, and subjected to molecular-dynamics simulation with PMEMD module of AMBER16 program[33] by a previously reported methodology[34]. The molecular-mechanics parameters were provided by AMBER ff14SB force field and general amber force field (gaff).

The SPOR-RlpA:**1** X-ray structure was used as a template to computationally model the complex of SPOR-RlpA with O-acetylated PG (SPOR-RlpA:OAc; acetyl group modification on NAM(1) and NAM(3)) and N-deacetylated PG (N-deacetylation on NAG(2) and NAG(4)). These PG modifications were modeled onto compound **1** in the SPOR-RlpA:**1** X-ray structure. There were no steric clashes, so that the sugars at position 1 to 4 remained as it was in the parent X-ray complex. The charges of the ligand atoms were calculated quantum-mechanically as described above and the complex was energy-minimized, equilibrated, and subjected to molecular-dynamic simulation using the same protocol as described above.

**Reporting summary.** Further information on research design is available in the Nature Research Reporting Summary linked to this article.

## Data availability
The atomic coordinates and structure factors of SPOR-RlpA, SPOR-RlpA:**1** (determined by co-crystallization and soaking experiments) and SPOR-RlpA:**3** complexes have been deposited in the Protein Data Bank under accession codes 6I05, 6I0A, 6I09 and 6I0N, respectively. A reporting summary for this Article is available as Supplementary Information file.

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

## Acknowledgements

The work in Spain was supported by grants from the Spanish Ministry of Science, Innovation and Universities (BFU2014-59389-P and BFU2017-90030-P to JAH) and in the USA by grants from the NIH (GM131685 and GM61629 to SM). D.A.D. is a Fellow of the Chemistry-Biochemistry-Biology Interface Program (NIH Training Grant T32GM075762) and a Fellow of the ECK Institute of Global Health at the University of Notre Dame. We thank Dr. Mayte Batuecas for kindly providing us with the cif files of compounds **1** and **3**. We thank the staff from ALBA synchrotron facility (Barcelona, Spain) for help during crystallographic data collection. We thank the Center for Research Computing of the University of Notre Dame for the computing resources.

## Author contributions

M.A. carried out crystallographic determinations and structural analysis. D.A.D., M.L., and B.B. performed the mass spectrometric experiments. T.D.G. performed crystallization experiments. D.A.D. performed the SEM experiments. S.D.B. cloned the genes and prepared the proteins. K.V.M. performed dynamics simulations. M.L. and D.H. prepared the synthetic ligands. T.D.G. performed crystallization experiments. E.L. supported crystallography by protein purification. D.L. supported cellular experiments. S.M. and J.A.H. directed the teams. M.A, S.M., and J.A.H. wrote the paper. All authors edited the manuscript.

## Competing interests

The authors declare no competing interests.
