## [Peer Review File · Nature Communications]

Reviewers' comments:

Reviewer #1 (Remarks to the Author):

Authors perform structural and binding studies of the SPOR domain of the lytic transglycosylase RlpA from *Pseudomonas aeruginosa*. They prove by mass spectrometry that SPOR-RlpA is able to bind only denuded PG glycans and that the presence of the peptide stem abrogates binding.

These findings are further rationalised using xray crystallography and Molecular Dynamics. The work is novel and well conducted, and the manuscript well written. Only a few typos should be fixed, e.g. "uniquely" instead of "unique" at p. 3, line 69.

I believe that this work adds significant novelty to the literature data since, for the first time, it provides clues on how a large family of SPOR domains (with low sequence identities) share a common pattern for recognition of septal PG. Indeed, recognition is proven to be dictated by PG or, better to say, by the action of amidases which deplete PG of peptide stems.

I definitely recommend publication of this work, after authors fix a few typos along the manuscript.

Reviewer #2 (Remarks to the Author):

Review Alcorlo et al., "Structural basis of denuded glycan recognition by SPOR domains in bacterial cell division", under consideration at Nature Communications 2019

In many bacteria, proteins containing a sporulation-related repeat (SPOR) domain play essential roles in remodeling the peptidoglycan cell wall during cell division. The manuscript by Alcorlo et al. describes the mechanism by which SPOR domains recognise septal peptidoglycan (PG) devoid of peptide stems, so-called 'denuded PG'. They employ mass spectrometry to demonstrate specific binding of the SPOR domain of a conserved lytic transglycosylase, SPOR-RlpA, to synthetic derivatives of denuded PG. High-resolution X-ray structural analyses and sequence alignments allow for the identification of amino acid residues important for SPOR domains to bind septal PG. Furthermore, molecular dynamics simulations are performed to reveal structural dynamics of the SPOR domain when engaging PG derivatives with distinct chemical modifications.

The manuscript is well written, and almost all experiments seem to have been performed well. Novelty of the present work is mildly compromised by three existing structures of SPOR domains (references 13-15, one is ours I should declare) and maybe more so by mutagenesis studies that uncovered functionally important regions of the SPOR domain (references 14 and 26). However it should be noted that use of the various glycan derivatives is a real step forward and the work provides important insights that are definitive (no conformational change upon binding, binding of two PG glycan repeats only and the reasoning why only denuded glycans bind). It is a big step for people interested in bacteria cell division and cell wall synthesis, but probably of somewhat limited interest to others. So, this reviewer feels that it is probably a borderline case for Nat Comms but overall a good study.

Major comments

1) Surprisingly, the crystal structures show statistics that might indicate a lack a care during refinement. Structure validation reports show that the bound glycan ligand in each of three SPOR-RlpA-glycan complex structures has a substantial portion of bond length and bond angle outliers. This indicates that many monosaccharides (e.g., AMU, AMV, AHO, and NAG) deviate from the standard chair conformation. Furthermore, despite being determined at better than 1.5 Å

resolution, these three structures show a high clashscore (e.g., a score of 9 for 6I09) and a non-negligible portion of side chain rotamer outliers (e.g., 8% for 6I0N, could just be the ones not defined by density?). These observations raise mild concerns about the quality of these structures and maybe even some structural interpretations presented in the manuscript. The authors should have a critical look and possibly re-refine these three structures and validate all glycan structures using the program Privateer (Agirre et al, Nat Struct Mol Biol. 2015, 22, 833–834). Please also provide a description of detailed model refinement procedures (e.g., restraints used for protein and ligand) in Materials and Methods.

2) The present work identifies conserved molecular determinants of denuded glycan recognition by SPOR domains based upon sequence alignments and structural comparisons.

a) These results are not fully discussed in the context of previous functional studies. In particular, the authors should include in the subsection "Structural principles for glycan-chain recognition by SPOR domains" a discussion about the results presented in reference 26. In this report, a number of residues including those identified based on current structural analysis were shown to be important for the functions of the SPOR domain in FtsN.

b) These structural observations are not validated by mutagenesis studies in the current work. The authors could investigate how mutations on the previously undescribed basic patch (R302, R309 and R311) and at position 274 affect binding of SPOR-RlpA to synthetic PG derivatives (or isolated PG), septal localisation of RlpA, and cell division. Although mostly loss-of-function type experiments, they could still help to validate some of the assertions.

3) Lines 137-143. The authors described the relative binding affinity of four ligands to SPOR-RlpA. However, these statements were not supported by quantitative measurements. It is stated that binding curves could not be generated because of a lack of material for the glycan derivatives. However, there are methods that consume very little material while providing excellent measurements of affinities and sometimes also rates, such as SPR, Octet, BLI, thermophoresis, and others. It would strengthen the entire argument quite a bit as so much of the discussion rests on what does and what does not.

4) As an experimentalist it was not entirely clear to me what the MD simulations added. Without further verification of the 'discovered' mechanisms it seems a bit fanciful and I think could even be removed?

Minor comments

1) Line 189. Please describe what the calculation of RMSD is based on, e.g., equivalent Ca atoms, main chain atoms, or all atoms.

2) Line 373. The statement that "N-deacetylated PG is not likely to be recognised by SPOR-RlpA" is not supported by experimental data. Please reword.

3) Line 430. Please clarify if sequence identity refers to amino acid sequence or nucleotide sequence.

4) For readers who are not familiar with bacterial cell division, a general introduction to the septal localisation and functions of SPOR domain-containing proteins (particularly lytic transglycosylases) in cell division would be informative. Also, Fig 7bc could be at the beginning, part of the introduction and not discussion as currently. Figure 7b is not great, showing a horribly de-hydrated (dead) bacterium prepared for SEM.

5) Given that the denuded glycans are transiently present during septal PG synthesis, please include a discussion about how the current work helps to understand how SPOR domains engage a transient substrate. Since the denuded glycans are degraded by lytic transglycosylases, it would also be helpful to discuss how the SPOR domain and the catalytic domain of RlpA might function in a coordinated manner.

6) The methods are generally detailed. However, below are a few items to clarify.

a) Supporting information Page 2 Paragraph 2. Please state the incubation condition for forming SPOR-RlpA-ligand complexes. The authors should also explain the rationale of using a ligand concentration of 500 μM for all synthetic PG derivatives in mass spectrometry analysis. For instance, the binding stoichiometry of SPOR-RlpA with compound 4 shown in Fig. 2 probably depends on ligand concentration.

b) Supporting information Page 4 Paragraph 2. Please provide more details about the phasing procedures. For instance, what search fragments were used? Were the ligand-bound structures determined similarly using de novo phasing or molecular replacement?

c) Please provide a description of the criteria (e.g., within a given radius to any atom in the ligand) by which amino acid residues interacting with a bound ligand are identified. An example is the amino acids shown in Fig. 3.

7) Corrections/clarifications are suggested for the following display items.

a) Fig. 2. Please describe what Y axis represents and the scale of signal intensity for each panel. It would be clearer to state the molecular mass for each ligand in the figure legend.

b) Fig. 3 and Supplementary Fig. 5. In Fig. 3B, the left portion of the funnel-like cavity is almost entirely positively charged, extending well beyond the basic patch formed by R302, R309 and R311. Is this caused by the use of a narrow electrostatic scale? Please include a color key to show the scale of surface electrostatic potential. Similarly, it would be helpful to include a color key that illustrates the scale of B factor in Supplementary Fig. 8.

c) Fig. 5. Please describe in the figure legend how the dashed line was generated.

d) Fig. 6B and Supplementary Fig. 9. Please provide a definition for each of the interatomic distances expect for Q270-COO and Q270-N.

e) Supplementary Table 1.

i) Please include R_{merge} (or R_{meas}) as well as average B-factor for all atoms, protein atoms, water, and ligand atoms.

ii) The equation for R_{pim} seems incorrect in that $1/2$ should be square root I think.

iii) The structure of SPOR-RlpA:3 has a large bond length r.m.s deviation of 0.036 Å. Is this a typo?

iv) For the same type of values (e.g., wavelength and $\langle I/\sigma(I) \rangle$), please use the same number of digits after the decimal point consistently.

f) Supplementary Fig. 2. Please include a scale bar in Panel B. It is not clear from Panel C that SPOR-RlpA crystals diffracted to 1.20 Å resolution. Please include a close-up view of reflections around 1.32 Å resolution and beyond.

g) Supplementary Fig. 3. Panel A, please use a close-up view of the map in which readers can appreciate typical electron densities at 1.20 Å resolution. In Panels B and C, one could also show $F_o - F_c$ omit map or $F_{\text{complex}} - F_{\text{apo}}$ (if isomorphous) difference map for the bound ligand. Please also discuss what accounts for lower quality ligand density than that of the surrounding amino acid residues. Occupancy issues?

Reviewer #3 (Remarks to the Author):

SPOR domains are small peptidoglycan binding domains found in a large number of bacterial proteins, many of them important for cell division. Previous studies have established that (i) SPOR domains have a conserved fold consisting of a 4-stranded anti-parallel beta sheet and two alpha helices, (ii) the PG binding site is probably the concave face of the beta sheet, and (iii) the PG ligand is a “denuded” glycan, meaning a stretch of NAG-NAM devoid of stem peptides. Nevertheless, until now there has been no structure of a SPOR:PG complex, so it has not been possible to understand the molecular details of the SPOR:glycan interaction.

The manuscript under review fills this knowledge gap by providing high quality crystal structures of the SPOR domain from the *Pseudomonas* RlpA protein both alone and in complex with several synthetic PG glycan ligands. The significance of these findings is enhanced by using computational techniques to model how the glycan probably binds to three other SPOR domains for which structures have been solved in the absence of a ligand. For two of these other SPOR domains (DamX, FtsN) site-directed mutagenesis has been undertaken to identify amino acids important for PG-binding, although in many cases septal localization was assayed as a proxy for PG binding.

As expected based on structure-function studies for DamX SPOR and FtsN SPOR, the glycan binds at the beta-sheet of RlpA SPOR and interacts with amino acids previously implicated in glycan binding in the other SPOR domains. But now we can see for the first time a wealth of detail that genetics could never provide. One of the more interesting findings is that many of the important protein-ligand interactions involve main-chain atoms rather than amino acid R groups. This nicely explains how multiple SPOR domains with very little sequence conservation can nevertheless bind to the same glycan ligand. The structures also reveal for the first time the chemical features of the glycan that are important for binding. Thus, we can now understand why only denuded glycans bind to the domain; it turns out that peptide side chains sterically block binding and their removal exposes charged lactyl carboxylates that interact with a conserved glutamine identified in previous studies as the most important residue for binding (but without any understanding of why that Q was important for binding).

Overall these new findings constitute an important advance that will be of interest to structural biologists, microbiologists and scientists interested in developing new antibiotics.

Major concerns

1. The authors did not test their binding model by site-directed mutagenesis of the amino acids they say are important for binding the glycan. In some cases this is understandable. Residues corresponding to Q270 and A273 have been shown to be important in DamX SPOR and FtsN SPOR, so there is no need to replicate that for RlpA SPOR. And backbone contacts are not very amenable to probing by mutagenesis. But some of the interactions seen in the structure are new and need to be tested, especially because the authors use language that ascribes importance to these new interactions. The basic patch formed by R302, R309 and R311 is described as “critical for binding the carboxylate and acetyl groups of NAM1” (lines 220-221). Likewise, the aromatic ring F274 is said to make a “key contact” with NAM3 (line 331). If true, then mutating the residues in question will have a large effect on localization and/or PG binding. I think the authors should do that. In view of the very large number of contacts the protein makes with the glycan it is plausible that substitutions at these sites will not actually have much impact on binding, in which case it would not be true they are “critical” or make “key contacts.”

2. The text does not put the new findings in proper context because relevant studies of other SPOR domains are not summarized accurately or in sufficient detail as to be informative. In particular, site-directed mutagenesis of the SPOR domains from DamX and FtsN led the Weiss lab to propose years ago that the PG-binding site is the beta-sheet (e.g., see Figure 4 of Duncan et al., 2013).

Yet someone reading the manuscript under review would have no idea that (a) mutagenesis had implicated the beta-sheet in PG binding and (b) that is exactly how those studies were interpreted at the time. But issues related to SPOR domain mutagenesis are only the tip of the iceberg when it comes to the pervasive failure to properly acknowledge prior work.

Specific comments

1. Line 81 says 3 of the 4 *E. coli* SPOR proteins are indispensable, but only 1 is essential (FtsN).
2. Line 97-98. This passage should cite Jorgenson et al., currently ref 21 in the manuscript, the first paper to show LT activity.
3. Line 102. The sentence mixes and matches *E. coli* and *P. aeruginosa* findings in ways that are inaccurate. RlpA localization has been shown in *E. coli* (refs 9 and 10) and in *P. aeruginosa* (Ref. 21). It is required for efficient daughter cell separation and rod shape in *Pseudomonas* (ref 21) but not in *E. coli* (refs 9 and 10). Indeed, whether *E. coli* RlpA is even an LT is an open question.
4. The introduction is not well tailored to the experimental content of the paper. It focuses on the biological roles of SPOR domain proteins in bacterial cell division without discussing previous structure-function studies. But this is a paper about how SPOR domains bind PG, so the introduction should describe the conserved RNP-fold and note that PG is proposed to bind at the concave face of the beta-sheet. The introduction should also explain that this hypothesis is based on site-directed mutagenesis of DamX SPOR and FtsN SPOR, which revealed that residues important for septal localization and PG-binding are in the exposed face of the beta sheet. The relevant mutagenesis, localization and PG binding assays were published in Williams et al. 2013, Duncan et al., 2013, and Yahashiri et al., 2015. Having provided this background, it would then make sense to explain that previous studies could not provide detailed insight into the SPOR:glycan interaction or explain why stem peptides interfere with binding. These are new insights that come from the manuscript under review.
5. Line 137. I like the mass spectrometry assay. It's a clever way to study binding when little substrate is available. But I wonder if the authors have any rationale for why the tetrasaccharide binds better than the octasaccharide? It seems to me that the octasaccharide contains within it 3 tetrasaccharide binding sites and thus should bind better for statistical reasons alone.
6. Line 149. I stumbled over the claim that this is the first direct evidence for binding naked glycans. It seems to me that the evidence in Yahashiri et al., 2015, is pretty clear and direct. But I agree that the mass spec assay is even more direct. I guess I would say this is the first time binding has been demonstrated with a homogenous, chemically-defined ligand. In any case, the claim of priority is sufficiently murky that it requires context.
7. Line 198. The failure of the ligand to bind to the convex face of the SPOR domain is also consistent with mutagenesis of FtsN, in which essentially every surface exposed amino acid on that side of the domain was altered without finding any noteworthy effects on septal localization. That mutagenesis was undertaken with the express purpose of looking for binding sites outside of the beta-sheet, which had already been implicated in binding PG in a study of DamX SPOR. The FtsN SPOR mutagenesis data strengthen the argument that the only glycan binding site is the beta sheet, and the authors should note it.
8. Line 220. As noted above, the existence of the basic patch is clear enough but I question whether it is "critical" and in any case see no data that speak for or against this point.
9. Line 125-127. "attempted to tackle" makes it sound like the previous studies were a failure. Also, please add PG binding assays to the list and include the Yahashiri et al. 2015 paper among the citations. The mutants were studied by more than just in vivo localization!

10. Line 312. As noted above, it's unclear to me why the contact formed by F274 is considered "key." I think the authors have to characterize a mutant if they want to make such statements.

11. Line 376-414. Here the authors generalize their findings with RlpA SPOR by using molecular modeling to propose a more general model for the SPOR:glycan interaction. This is good because it broadens the significance of the report. But the section would be more useful if the authors pointed out which of the proposed interactions have experimental support and which do not. Such a discussion is all the more important because the authors introduce the section by claiming it will provide a structural understanding for previous mutagenesis studies of DamX SPOR and FtsN SPOR. I had to spend a couple of hours with the relevant reports in one hand and the models (Supplemental Figure 10) in the other to figure out which of the many contacts in the figure had been tested experimentally and what the outcome was. That's too much work, and in the end it is the readers rather than the authors who are figuring out how the structure explains the functional studies.

12. Line 390. The text indicates K418 of DamX is part of the basic patch while supplemental figure 10 shows a K414. There seems to be a typo here. Residue 414 of DamX is an N. There are K's at 413 and 418.

13. Line 412. Here the authors propose based on their model that W416 of DamX SPOR and W283 of FtsN SPOR interact with glycans. Indeed, DamX W416 contributes to septal localization and PG binding (Williams et al., 2013). FtsN W283 contributes to septal localization but PG binding was not tested (Duncan, 2013). This should be noted. In other words, the passage reads as if the authors have identified important residues that were previously overlooked when in reality it had already been shown that W416 and W283 were important. What's new is how they might interact with the glycan.

14. Line 413. Typo: Should this be Supplementary Figure 10 rather than 9?

15. RlpA F274, which is proposed to make pi interactions with NAM3, is not discussed in the generalized model. But I see that F274 is aligned with F255 of FtsN SPOR in Supplemental Figure 10, suggesting the authors think F255 might also make pi interactions with NAM3. Based on visual inspection of FtsN SPOR in PyMOL, I think the aromatic ring of F255 is completely buried and therefore not available to interact with the glycan. Please double check and revise the figure accordingly if warranted. Maybe I am mistaken.

16. Line 446. Typo: delete "collective the process"

17. Conclusions (lines 428-479). This section is problematic on multiple levels. Only the first and last paragraphs actually pertain to insights or advances that stem from the study under review. I suggest keeping these two paragraphs and deleting the intervening two paragraphs, which summarize ideas lifted from other investigators, for most part without proper attribution. Specific concerns follow.

18. Lines 447-455. These sentences rehash de Boer's 2009 model for how PG hydrolases work together to regulate recruitment of SPOR proteins to the divisome without citing him. There is no new information in the manuscript under review that pertains to the model, and no new suggestions in the passage, so I suggest dropping it. If it is to be kept the passage needs to be revised to make it clear that these are not original insights. The basic model was first published in Gerding et al., 2009. Noteworthy extensions are in Busiek & Margolin, 2014, who showed FtsN also has a cytoplasmic localization signal that complicates the story, and in Yahashiri et al., 2015, who provided experimental support for the role of LTs in releasing SPOR proteins from septal PG. A review from Weiss's lab that provides a more nuanced treatment of the model and discusses additional reasons why using a SPOR domain to drive localization might be beneficial (Yahashiri et

al., 2017).

19. Lines 456-455. This passage points out that the SPOR domain of RlpA delivers the enzyme to its substrate in the septum. There is a new finding here because the authors show for the first time that the SPOR binding site is a tetrasaccharide. But the notion that the SPOR domain is a clever way to deliver RlpA to its substrate is already in Jorgenson et al., 2014, and in Yahashiri et al., 2017. The priority of these papers should be credited.

20. Figure 7. It should probably be noted that the cartoon in 7B is adapted from Gray et al., 2015. The cartoon in 7C illustrating sequential activity of amidases followed by RlpA is nice, and the point that the SPOR domains binds a tetrasaccharide is an original insight, but the model was first put forth by Jorgenson et al., 2014. Their priority should be noted.

21. Line 466-468. This passage asserts that all previous localization studies were done in living cells and therefore it has not been possible until now to exclude that SPOR domains localize to the septum by binding a protein rather than a PG structure. This is not correct. Yahashiri et al., 2015, demonstrated localization on purified PG sacculi, which (a) is not in a living cell, and (b) pretty much ruled out the potential involvement of a septal protein.

22. Line 468-470. Lines 468-470 read "The work that we disclose herein provides the first clear direct evidence that SPOR-RlpA binds denuded PG to the exclusion of PG with peptide stems." Claims of priority like this are always problematic because new findings build on previous work and what constitutes "clear evidence" to some might not be so unambiguous to others. In the manuscript under review, the authors document binding of a purified SPOR domain to chemically defined PG ligands, showing that a denuded glycan binds but a glycan with stem peptides does not. These experiments go beyond what has been demonstrated before. Nevertheless, I do not think the issue was up for grabs because the previous data were clear and direct enough. In particular, Yahashiri et al. showed that purified SPOR domains only bind to PG sacculi that contain denuded glycans. Their data included showing that amidase treatment of PG increased binding, while selective removal of denuded glycans with RlpA abrogated binding. If the authors think those experiments come with important caveats they should spell out the credible alternative interpretation(s) that their new findings now exclude. But I think the important new contribution here is explaining the requirement for a denuded glycan more so than showing such a requirement in the first place.

23. Line 471. Here the authors seem to be taking credit for providing evidence denuded glycans are enriched in septal PG. But priority again goes to Yahashiri et al., 2014.

24. Supplemental Figure 10. This figure presents models for how a glycan ligand binds to SPOR domains from CwIC, FtsN and DamX. As noted elsewhere, some of the modeled interactions have experimental support from previous site-directed mutagenesis. Although the authors allude to those studies in the general sense, somewhere in the paper they need to address the concordance between the experimental data and their models in detail. Also, should DamX K414 be K413?

Point by point answer to the reviewers. Please find the requested points below

Reviewers' comments:

Reviewer #1 (Remarks to the Author):

Authors perform structural and binding studies of the SPOR domain of the lytic transglycosylase RlpA from *Pseudomonas aeruginosa*. They prove by mass spectrometry that SPOR-RlpA is able to bind only denuded PG glycans and that the presence of the peptide stem abrogates binding.

These findings are further rationalised using xray crystallography and Molecular Dynamics. The work is novel and well conducted, and the manuscript well written. Only a few typos should be fixed, e.g. "uniquely" instead of "unique" at p. 3, line 69.

I believe that this work adds significant novelty to the literature data since, for the first time, it provides clues on how a large family of SPOR domains (with low sequence identities) share a common pattern for recognition of septal PG. Indeed, recognition is proven to be dictated by PG or, better to say, by the action of amidases which deplete PG of peptide stems.

I definitely recommend publication of this work, after authors fix a few typos along the manuscript:

We are grateful to the Reviewer#1 for her/his positive comments. We have corrected the indicated typos. The manuscript was read thoroughly for other typos during the revision.

Reviewer #2 (Remarks to the Author):

Review Alcorlo *et al.*, "Structural basis of denuded glycan recognition by SPOR domains in bacterial cell division", under consideration at Nature Communications 2019

In many bacteria, proteins containing a sporulation-related repeat (SPOR) domain play essential roles in remodeling the peptidoglycan cell wall during cell division. The manuscript by Alcorlo *et al.* describes the mechanism by which SPOR domains recognise septal peptidoglycan (PG) devoid of peptide stems, so-called 'denuded PG'. They employ mass spectrometry to demonstrate specific binding of the SPOR domain of a conserved lytic transglycosylase, SPOR-RlpA, to synthetic derivatives of denuded PG. High-resolution X-ray structural analyses and sequence alignments allow for the identification of amino acid residues important for SPOR domains to bind septal PG. Furthermore, molecular dynamics simulations are performed to reveal structural dynamics of the SPOR domain when engaging PG derivatives with distinct chemical modifications.

The manuscript is well written, and almost all experiments seem to have been performed well. Novelty of the present work is mildly compromised by three existing structures of SPOR domains (references 13-15, one is ours I should declare) and maybe more so by mutagenesis studies that uncovered functionally important regions of the SPOR domain (references 14 and 26). However it should be noted that use of the various glycan derivatives is a real step forward and the work provides important insights that are definitive (no conformational change upon binding, binding of two PG glycan repeats only and the reasoning why only denuded glycans bind). It is a big step for people interested in bacteria cell division and cell wall synthesis, but probably of somewhat limited interest to others. So, this reviewer feels that it is probably a borderline case for *Nat Comms* but overall a good study:

We thank Reviewer#2 for the laudatory comments. We agree with the comments on the several "firsts" that have been accomplished in the manuscript, with "definitive" results, as the reviewer

kindly points out. We hope that you agree with us that this multidisciplinary project is suited for the readership of Nat. Comms.

Major comments

1) Surprisingly, the crystal structures show statistics that might indicate a lack a care during refinement. Structure validation reports show that the bound glycan ligand in each of three SPOR-RlpA-glycan complex structures has a substantial portion of bond length and bond angle outliers. This indicates that many monosaccharides (e.g., AMU, AMV, AHO, and NAG) deviate from the standard chair conformation. Furthermore, despite being determined at better than 1.5 Å resolution, these three structures show a high clashscore (e.g, a score of 9 for 6I09) and a non-negligible portion of side chain rotamer outliers (e.g., 8% for 6I0N, could just be the ones not defined by density?). These observations raise mild concerns about the quality of these structures and maybe even some structural interpretations presented in the manuscript. The authors should have a critical look and possibly re-refine these three structures and validate all glycan structures using the program Privateer (*Agirre et al, Nat Struct Mol Biol. 2015, 22, 833–834*). Please also provide a description of detailed model refinement procedures (e.g., restraints used for protein and ligand) in Materials and Methods:

We thank Reviewer#2 for careful analysis of the structures. We have followed the Reviewer instructions and we have re-refined the structures with bound glycan ligands. The re-refined structures present now a much lower clashscore (1.48 vs 9 for 6I09, 2.90 vs 8 for 6I0A and 2.83 vs 7 for 6I0N) and a negligible portion of side-chain rotamer outliers. After re-refinement of the structures, all glycan structures were validated with the program Privateer using a mask radius around the sugar atoms of 1.5 Å, per reviewer's kind suggestion. In all cases, no issues were found after running the program, including 0 stereochemical problems, 0 unphysical puckering amplitude and 0 unlikely ring conformation. This validation was further corroborated when the re-refined structures were validated at the PDB protein data bank. In the validation reports, in the section concerning ligand geometry, we got the following:

- There are no bond length outliers.

-There are no bond angle outliers.

-There are no chirality outliers.

-There are no ring outliers.

Information on the refinement procedures is now included in Material and Methods section. While the quality of the structures have been improved after re-refinement—for which we are grateful to the reviewer—it is important to state that no significant differences were found when compared with our previous structures. This indicates that our structural interpretations presented in the manuscript were not compromised.

2) The present work identifies conserved molecular determinants of denuded glycan recognition by SPOR domains based upon sequence alignments and structural comparisons.

a) These results are not fully discussed in the context of previous functional studies. In particular, the authors should include in the subsection “Structural principles for glycan-chain recognition by SPOR domains” a discussion about the results presented in reference 26. In this report, a number of residues including those identified based on current structural analysis were shown to be important for the functions of the SPOR domain in FtsN:

Reviewer#3 also suggested this and we were very pleased to expand the subsection “Structural principles for glycan-chain recognition by SPOR domains” in order to detail and put in context our present results with previously reported works on SPOR-FtsN and SPOR-DamX. We think that the revised version clearly presents the previous work done. We have also included a new Supplementary Table (Supp Table 3) to identify the residues that could be involved in denuded

glycan recognition in the SPOR domains of CwIC, FtsN and DamX, based on the SPOR-RlpA complexes. The residues that were previously identified by mutagenesis in FtsN and DamX studies are highlighted in the Table.

b) These structural observations are not validated by mutagenesis studies in the current work. The authors could investigate how mutations on the previously undescribed basic patch (R302, R309 and R311) and at position 274 affect binding of SPOR-RlpA to synthetic PG derivatives (or isolated PG), sepal localisation of RlpA, and cell division. Although mostly loss-of-function type experiments, they could still help to validate some of the assertions:

We have now analyzed by mutagenesis and mass spectrometry how the basic patch (R302, R309 and R311) affects binding of SPOR-RlpA to our synthetic PG derivative. We did alanine scanning at each (R302A, R309A and R311A) site and then we prepared the triple mutant (where each arginine was replaced by alanine). Although each single alanine substitution had an effect on binding to peptidoglycan, the cluster of three arginines serves like a cushion in recognition of the ligand. The largest effect was seen for the triple mutant. The genes for two additional mutants, Q270A (included as a control) and F274A, were successfully cloned and expressed. However, the solubility of the resultant protein was dramatically altered, requiring the presence of detergents. This precluded the ability to test binding of the proteins with the peptidoglycan ligand by mass spectrometry. Our results with the four variants of SPOR-RlpA (Figure 7) further support the important role of the basic patch on glycan recognition.

We decided to further explore the effect of the mutations by in vivo localization studies. However, we find problems in overexpression and/or in clear septal localization for the positive control, despite the three different approaches we followed: (i) Tat(E. coli)-GFP-SPOR-RlpA on E. coli, (ii) Tat(P. aeruginosa)-Cherry- SPOR-RlpA on P. aeruginosa and (iii) RlpA(P. aeruginosa)-Cherry on P. aeruginosa Δ RlpA.

For this, we used the pSEVA234 plasmid (Silva-Rocha et al Nucleic Acids Res. 2013 Jan; 41: D666–D675), for E. coli and P. aeruginosa protein expression. Three DNA fragments corresponding to a periplasm exportation signal, rlpA and gfp were fused to a single in-frame DNA fragment using joining PCR and the fragment was cloned in pSEVA234 plasmid under the expression control of trc IPTG-inducible promoter. We designed our construct trying to reproduce the methodology exposed in Yahashiri A et al., (PNAS 2015 8;112(36):11347-52), therefore we fused the twin-arginine signal peptide (Tat) of TMAO reductase (TorA) at the GFP N-terminus (Thomas JD et al., Mol Microbiol. 2001 39(1):47-53) and the SPOR-RlpA domain at the GFP C-terminus. All constructs were confirmed by DNA sequencing. In a parallel approach, we used joining PCR to generate constructs for P. aeruginosa. In this case, we chose the periplasm exportation signal of protein PlcH, which also depends upon a functional Tat system and a Tat signal sequence. We used joining PCR to generate a single fragment of the Tat exportation signal, rlpA and cherry (red fluorescence protein).

We did not detect clear septal localization of the wt GFP-SPOR-RlpA fusion (positive control) in E. coli. We tried different conditions to visualize our TAT-GFP-SPOR-RlpA fusion including temperature (28°C, 30°C and 37°C), rich (LB) or minimal medium, different concentrations of IPTG (inducer) and induction times. We detected the fluorescence in the periplasm, indicating a productive exportation process. Many cells showed fluorescence at the cell poles, probably reflecting misfolding issues. The constructs that were assayed in P. aeruginosa showed a low fluorescence signal. It is possible that the Cherry protein does not show a signal as intense as GFP, nonetheless we could not use GFP in these experiments because P aeruginosa shows intrinsic green fluorescence. From this signal, pattern, it was not possible to detect fluorescence signal associated with cell-division septa. We then, produced the constructs in P. aeruginosa. For this, we chose the signal found in the gene coding for protein PlcH, which also depends upon a functional Tat system and a Tat signal sequence. RlpA was fused to a red fluorescence signal (RlpA-Cherry) and expressed in a P aeruginosa Δ rlpA mutant. In this strain, the red signal does not show clear septal localization. It is possible that the intrinsic fluorescence of P. aeruginosa still affects our fluorescence detection limit, yet we have not been able to reproduce the GFP-SPOR-RlpA localization pattern exposed in Yahashiri A et al., 2015.

It is reported that RlpA shows a rather complex subcellular distribution pattern, with the accumulation in the cell-division septum as well as in discrete foci in the cellular envelope (Yahashiri A et al., 2015, GFP-SPOR-RlpA). It is thus likely that the organization pattern of RlpA is more complex than previously appreciated and that our experimental scenario is not optimal to visualize subtle differences in the organization pattern. We hope that the editor and the reviewer would appreciate the amount of work and effort that we put forth for these experiments.

3) Lines 137-143. The authors described the relative binding affinity of four ligands to SPOR-RlpA. However, these statements were not supported by quantitative measurements. It is stated that binding curves could not be generated because of a lack of material for the glycan derivatives. However, there are methods that consume very little material while providing excellent measurements of affinities and sometimes also rates, such as SPR, Octet, BLI, thermophoresis, and others. It would strengthen the entire argument quite a bit as so much of the discussion rests on what does and what does not:

Well, there is nothing to disagree with the reviewer's comments. We routinely use these methods ourselves in our various projects. However, the several ligands that we have used in this study each has been prepared in >20 non-trivial synthetic steps. There is simply not enough of these ligands to do the quantitative studies that are requested. Please bear in mind that the conditions have to be explored over a range of concentrations and then the studies must be replicated. We used mass spectrometry, which uses small amounts of sample, under the identical conditions for all experiments. So, the set of experiments can be compared to each other to get a sense of affinities, though the actual dissociation constants could not be had. This is a limitation for us (and frankly for anyone else who would do the same kind of experiments). We note that the fact that we have prepared the small quantities of the valuable (and rare) ligands for these studies indeed differentiates our work from the earlier reports. We hope that the reviewer appreciates our situation.

4) As an experimentalist it was not entirely clear to me what the MD simulations added. Without further verification of the 'discovered' mechanisms it seems a bit fanciful and I think could even be removed?

We respectfully disagree. We believe (and Reviewer#3 concurs) that this section is an important part of the manuscript. The MD simulations on SPOR-RlpA:1 and SPOR-RlpA:3 complexes provide information on stability of the complexes as a function of time. These simulations beautifully dovetail with the X-ray structures on which they are based. Furthermore, the X-ray structures plus the simulations, were instrumental in assessing the importance and the consequences of the natural PG modifications, which are not easily assessed by experiments (for lack of the requisite PG fragments).

Minor comments

1) Line 189. Please describe what the calculation of RMSD is based on, e.g., equivalent C α atoms, main chain atoms, or all atoms:

The RMSD is calculated based on equivalent C α atoms. This has been indicated in the main text of the manuscript.

2) Line 373. The statement that "N-deacetylated PG is not likely to be recognised by SPOR-RlpA" is not supported by experimental data. Please reword:

The sentence has been rewritten as follows: "... according to the simulation, N-deacetylated PG is not likely to be recognized by SPOR-RlpA."

3) Line 430. Please clarify if sequence identity refers to amino acid sequence or nucleotide sequence:

It refers to amino acid sequence and it has been indicated in the text.

4) For readers who are not familiar with bacterial cell division, a general introduction to the septal localisation and functions of SPOR domain-containing proteins (particularly lytic transglycosylases) in cell division would be informative:

We have included the requested information by adding text (highlighted in yellow on pages 3 and 4) on the revised version.

Also, Fig 7bc could be at the beginning, part of the introduction and not discussion as currently: *According to the Reviewer suggestion we have moved Figure 7B and 7C panels to the Introduction section that now belong to Figure 1.*

Figure 7b is not great, showing a horribly de-hydrated (dead) bacterium prepared for SEM:

Fig. 7B shows how the typical surface of Pseudomonas strain PA01 is envision under the microscope. We see this type of surface, a representation of the top of the O-antigen in the surface clusters of lipid A, routinely. I draw your attention to the fact that this image has no informational content for our manuscript beyond merely showing two bacteria in the process of septation/separation. If you feel that we need to take this panel out, we will do so. However, we would like to keep it.

5) Given that the denuded glycans are transiently present during septal PG synthesis, please include a discussion about how the current work helps to understand how SPOR domains engage a transient substrate:

To address this request, we have added new text in the Concluding Remarks (page 24, lines 580-586).

Since the denuded glycans are degraded by lytic transglycosylases, it would also be helpful to discuss how the SPOR domain and the catalytic domain of RlpA might function in a coordinated manner:

We have added a statement on this matter to the Concluding Remarks (page 24, lines 573-578)

6) The methods are generally detailed. However, below are a few items to clarify.

a) Supporting information Page 2 Paragraph 2. Please state the incubation condition for forming SPOR_{-RlpA}-ligand complexes:

It was 30 min. Added to the text (page 25, lines 624).

The authors should also explain the rationale of using a ligand concentration of 500 μ M for all synthetic PG derivatives in mass spectrometry analysis. For instance, the binding stoichiometry of SPOR_{-RlpA} with compound 4 shown in Fig. 2 probably depends on ligand concentration.

An excess of ligand was used to saturate the binding site and allow the detection of weak interactions. Added to the text (page 25, lines 626-627).

b) Supporting information Page 4 Paragraph 2. Please provide more details about the phasing procedures. For instance, what search fragments were used?:

Included on revision (page 27, lines 688-689).

Were the ligand-bound structures determined similarly using de novo phasing or molecular replacement?:

By molecular replacement of the apo structure. This point has been clarified in the Methods section (page 27, lines 692-693).

c) Please provide a description of the criteria (e.g., within a given radius to any atom in the ligand) by which amino acid residues interacting with a bound ligand are identified. An example is the amino acids shown in Fig. 3:

Interactions with bound ligand were analyzed using LigPlot⁺ default parameters that are now included in the legend of Supplementary Figure 6 (SI page 10, lines 314-318).

7) Corrections/clarifications are suggested for the following display items.

a) Fig. 2. Please describe what Y axis represents and the scale of signal intensity for each panel. It would be clearer to state the molecular mass for each ligand in the figure legend:

Included on revision in the legend of Fig.3

b) Fig. 3 and Supplementary Fig. 5. In Fig. 3B, the left portion of the funnel-like cavity is almost entirely positively charged, extending well beyond the basic patch formed by R302, R309 and R311. Is this caused by the use of a narrow electrostatic scale? Please include a color key to show the scale of surface electrostatic potential. Similarly, it would be helpful to include a color key that illustrates the scale of B factor in Supplementary Fig. 8:

Done as requested.

c) Fig. 5. Please describe in the figure legend how the dashed line was generated:

Included on revision in the legend of Fig. 6.

d) Fig. 6B and Supplementary Fig. 9. Please provide a definition for each of the interatomic distances expect for Q270-COO and Q270-N:

Included on revision in the legend of Fig. 8.

e) Supplementary Table 1.

i) Please include Rmerge (or Rmeas) as well as average B-factor for all atoms, protein atoms, water, and ligand atoms:

The requested information has been included in the Supplementary Table (now Supplementary Table 2 in the revised version)

ii) The equation for Rpim seems incorrect in that 1/2 should be square root I think:

Reviewer is right; thanks. This has been corrected.

iii) The structure of SPOR-RlpA:3 has a large bond length r.m.s deviation of 0.036 Å. Is this a typo?:

Yes, it was. This has been corrected in the table.

iv) For the same type of values (e.g., wavelength and), please use the same number of digits after the decimal point consistently:

Done on revision (Supplementary Table 2).

f) Supplementary Fig. 2. Please include a scale bar in Panel B. It is not clear from Panel C that SPOR-RlpA crystals diffracted to 1.20 Å resolution. Please include a close-up view of reflections around 1.32 Å resolution and beyond:

We guess the Reviewer is referring to Supplementary Fig. 1. We have modified it accordingly. The image of the representative diffraction frame in panel C has been inverted to better show the reflections, now as white spots.

g) Supplementary Fig. 3. Panel A, please use a close-up view of the map in which readers can appreciate typical electron densities at 1.20 Å resolution. In Panels B and C, one could also show Fo-Fc omit map or Fcomplex-Fapo (if isomorphous) difference map for the bound ligand. Please also discuss what accounts for lower quality ligand density than that of the surrounding amino acid residues. Occupancy issues?:

We guess the Reviewer is referring to Supplementary Fig. 2. This figure has been modified according to the Reviewer request. Now, in panel A we have included a close-up view of the map in which typical electron densities at 1.2 Å resolution can be better appreciated. In panels B and C we have now included both the 2Fo-Fc and the Fo-Fc omit maps for both ligands. Regions in which ligand density is not so nicely defined are mobile and not in a fixed conformation as the rest of the molecule, rather that represent occupancy issues. This is in agreement with the higher B-factors observed for these regions, as shown in Supplementary Figure 8.

Reviewer #3 (Remarks to the Author):

SPOR domains are small peptidoglycan binding domains found in a large number of bacterial proteins, many of them important for cell division. Previous studies have established that (i) SPOR domains have a conserved fold consisting of a 4-stranded anti-parallel beta sheet and two alpha helices, (ii) the PG binding site is probably the concave face of the beta sheet, and (iii) the PG ligand is a “denuded” glycan, meaning a stretch of NAG-NAM devoid of stem peptides. Nevertheless, until now there has been no structure of a SPOR:PG complex, so it has not been possible to understand the molecular details of the SPOR:glycan interaction.

The manuscript under review fills this knowledge gap by providing high quality crystal structures of the SPOR domain from the *Pseudomonas* RlpA protein both alone and in complex with several synthetic PG glycan ligands. The significance of these findings is enhanced by using computational techniques to model how the glycan probably binds to three other SPOR domains for which structures have been solved in the absence of a ligand. For two of these other SPOR domains (DamX, FtsN) site-directed mutagenesis has been undertaken to identify amino acids important for PG-binding, although in many cases septal localization was assayed as a proxy for PG binding:

We thank Reviewer #3 for the precise analysis of the key findings of the manuscript.

As expected based on structure-function studies for DamX SPOR and FtsN SPOR, the glycan binds at the beta-sheet of RlpA SPOR and interacts with amino acids previously implicated in glycan binding in the other SPOR domains. But now we can see for the first time a wealth of detail that genetics could never provide. One of the more interesting findings is that many of the important protein-ligand interactions involve main-chain atoms rather than amino acid R groups. This nicely explains how multiple SPOR domains with very little sequence conservation can nevertheless bind to the same glycan ligand. The structures also reveal for the first time the chemical features of the glycan that are important for binding. Thus, we can now understand why only denuded glycans bind to the domain; it turns out that peptide side chains sterically block binding and their removal exposes charged lactyl carboxylates that interact with a conserved glutamine identified in previous studies as the most important residue for binding (but without any understanding of why that Q was important for binding).

Overall these new findings constitute an important advance that will be of interest to structural biologists, microbiologists and scientists interested in developing new antibiotics:

We are grateful for the laudatory and thoughtful comments.

Major concerns

1. The authors did not test their binding model by site-directed mutagenesis of the amino acids they say are important for binding the glycan. In some cases this is understandable. Residues corresponding to Q270 and A273 have been shown to be important in DamX SPOR and FtsN SPOR, so there is no need to replicate that for RlpA SPOR. And backbone contacts are not very amenable to probing by mutagenesis. But some of the interactions seen in the structure are new and need to be tested, especially because the authors use language that ascribes importance to these new interactions. The basic patch formed by R302, R309 and R311 is described as “critical for binding the carboxylate and acetyl groups of NAM1” (lines 220-221). Likewise, the aromatic ring F274 is said to make a “key contact” with NAM3 (line 331). If true, then mutating the residues in question will have a large effect on localization and/or PG binding. I think the authors should do that. In view of the very large number of contacts the protein makes with the glycan it is plausible that substitutions at these sites will not actually have much impact on binding, in which case it would not be true they are “critical” or make “key contacts.”:

Reviewer#2 also had commented on mutagenesis. Four different mutants were produced and their interaction with synthetic compound 1 measured by electrospray mass spectrometry. As observed in the new Figure 7 the relevance of the basic patch in substrate binding has been confirmed. The gene for F274A was successfully cloned and expressed, but unfortunately, the solubility of the resultant protein was dramatically altered, requiring the presence of detergents. This precluded the ability to test binding of the protein with the peptidoglycan ligand by mass spectrometry. Thus we have removed the word “critical” from the text when describing the role of F274.

2. The text does not put the new findings in proper context because relevant studies of other SPOR domains are not summarized accurately or in sufficient detail as to be informative. In particular, site-directed mutagenesis of the SPOR domains from DamX and FtsN led the Weiss lab to propose years ago that the PG-binding site is the beta-sheet (e.g., see Figure 4 of Duncan et al., 2013). Yet someone reading the manuscript under review would have no idea that (a) mutagenesis had implicated the beta-sheet in PG binding and (b) that is exactly how those studies were interpreted at the time. But issues related to SPOR domain mutagenesis are only the tip of the iceberg when it comes to the pervasive failure to properly acknowledge prior work:

We certainly did not intend to cause offence. We have now included a detailed description of the previous work on DamX and FtsN (see pages 21-22, lines 479-512). We have also included a new Supplementary Table (Supp Table 3) where the equivalent residues in RlpA, CwlC, FtsN and DamX involved in PG binding are indicated.

Specific comments

1. Line 81 says 3 of the 4 E. coli SPOR proteins are indispensable, but only 1 is essential (FtsN): *Thanks, this has been corrected on revision (page 3, line 82).*

2. Line 97-98. This passage should cite Jorgenson et al., currently ref 21 in the manuscript, the first paper to show LT activity:

Done.

3. Line 102. The sentence mixes and matches E. coli and P. aeruginosa findings in ways that are inaccurate. RlpA localization has been shown in E. coli (refs 9 and 10) and in P. aeruginosa (Ref. 21). It is required for efficient daughter cell separation and rod shape in Pseudomonas (ref 21) but not in E. coli (refs 9 and 10). Indeed, whether E. coli RlpA is even an LT is an open question:

The paragraph has been rewritten on revision (page 4, line 109-111).

4. The introduction is not well tailored to the experimental content of the paper. It focuses on the biological roles of SPOR domain proteins in bacterial cell division without discussing previous structure-function studies. But this is a paper about how SPOR domains bind PG, so the introduction should describe the conserved RNP-fold and note that PG is proposed to bind at the concave face of the beta-sheet. The introduction should also explain that this hypothesis is based on site-directed mutagenesis of DamX SPOR and FtsN SPOR, which revealed that residues important for septal localization and PG-binding are in the exposed face of the beta sheet. The relevant mutagenesis, localization and PG binding assays were published in Williams et al. 2013, Duncan et al., 2013, and Yahashiri et al., 2015. Having provided this background, it would then make sense to explain that previous studies could not provide detailed insight into the SPOR:glycan interaction or explain why stempeptides interfere with binding. These are new insights that come from the manuscript under review:

We agree. Additional text was introduced to address the comment (page 4, line 91-97).

5. Line 137. I like the mass spectrometry assay. It's a clever way to study binding when little substrate is available. But I wonder if the authors have any rationale for why the tetrasaccharide binds better than the octasaccharide? It seems to me that the octasaccharide contains within it 3 tetrasaccharide binding sites and thus should bind better for statistical reasons alone:

We reported the data as they are. It is true that the octasaccharide contains within it more potential binding sites than the tetrasaccharide. However, we have not detected more than one SPOR domain bound to the octasaccharide, indicating that in this complex some of the sugar units must be protruding from the ends of the complex. Why the tetrasaccharide bind better is not intuitively obvious to us at present, but they both bind.

6. Line 149. I stumbled over the claim that this is the first direct evidence for binding naked glycans. It seems to me that the evidence in Yahashiri et al., 2015, is pretty clear and direct. But I agree that the mass spec assay is even more direct. I guess I would say this is the first time binding has been demonstrated with a homogenous, chemically-defined ligand. In any case, the claim of priority is sufficiently murky that it requires context:

We have rewritten that sentence, with some text borrowed from the Reviewer's comment above, giving the Yahashiri et al. citation (page 7, lines 168-172).

7. Line 198. The failure of the ligand to bind to the convex face of the SPOR domain is also consistent with mutagenesis of FtsN, in which essentially every surface exposed amino acid on that side of the domain was altered without finding any noteworthy effects on septal localization. That mutagenesis was undertaken with the express purpose of looking for binding sites outside of the beta-sheet, which had already been implicated in binding PG in a study of DamX SPOR. The FtsN SPOR mutagenesis data strengthen the argument that the only glycan binding site is the beta sheet, and the authors should note:

We agree. The text is revised accordingly (page 10, lines 224-228).

8. Line 220. As noted above, the existence of the basic patch is clear enough but I question whether it is "critical" and in any case see no data that speak for or against this point:

The new mutagenesis and mass spectrometry experiments now provide further evidences on the relevant role of the basic patch (page 15, lines 326-344).

9. Line 125-127. "attempted to tackle" makes it sound like the previous studies were a failure. Also, please add PG binding assays to the list and include the Yahashiri et al. 2015 paper among the citations. The mutants were studied by more than just in vivo localization!:

That certainly was not our sentiment. We have rewritten the sentence (page 11, lines 256-258).

10. Line 312. As noted above, it's unclear to me why the contact formed by F274 is considered "key." I think the authors have to characterize a mutant if they want to make such statements:

Per comments in reply to Reviewer #2, we could not prepare the F274A mutant for analysis. We took out the descriptor out of the requisite text (page 17, line 372).

11. Line 376-414. Here the authors generalize their findings with RlpA SPOR by using molecular modeling to propose a more general model for the SPOR:glycan interaction. This is good because it broadens the significance of the report. But the section would be more useful if the authors pointed out which of the proposed interactions have experimental support and which do not. Such a discussion is all the more important because the authors introduce the section by claiming it will provide a structural understanding for previous mutagenesis studies of DamX SPOR and FtsN SPOR. I had to spend a couple of hours with the relevant reports in one hand and the models (Supplemental Figure 10) in the other to figure out which of the many contacts in the figure had been tested experimentally and what the outcome was. That's too much work, and in the end it is the readers rather than the authors who are figuring out how the structure explains the functional studies: The point is well taken. The reader should not search for the data nor for the mechanistic context.

We have included in the "Structural principles for glycan-chain recognition by SPOR domains" section an extensive description of the SPOR-FtsN and SPOR-DamX residues that provide experimental support to our structural model. As detailed before, we have also included a new Supplementary Table (sup. table 3) to help reader with this point.

12. Line 390. The text indicates K418 of DamX is part of the basic patch while supplemental figure 10 shows a K414. There seems to be a typo here. Residue 414 of DamX is an N. There are K's at 413 and 418:

Yes, it is K418 that is in the basic patch. The correct numbering of this residue has been introduced in the Supplementary Figure 10.

13. Line 412. Here the authors propose based on their model that W416 of DamX SPOR and W283 of FtsN SPOR interact with glycans. Indeed, DamX W416 contributes to septal localization and PG binding (Williams et al., 2013). FtsN W283 contributes to septal localization but PG binding was not tested (Duncan, 2013). This should be noted. In other words, the passage reads as if the authors have identified important residues that were previously overlooked when in reality it had already been shown that W416 and W283 were important. What's new is how they might interact with the glycan:

We agree. We have addressed this issue with the description that we provide in response to point 11 (together with new Supplementary Table 3).

14. Line 413. Typo: Should this be Supplementary Figure 10 rather than 9?

Yes. The typo is corrected on revision.

15. RlpA F274, which is proposed to make pi interactions with NAM3, is not discussed in the generalized model. But I see that F274 is aligned with F255 of FtsN SPOR in Supplemental Figure 10, suggesting the authors think F255 might also make pi interactions with NAM3. Based on visual inspection of FtsN SPOR in PyMOL, I think the aromatic ring of F255 is completely buried and therefore not available to interact with the glycan. Please double check and revise the figure accordingly if warranted. Maybe I am mistaken:

The reviewer is correct. Indeed we see that in FtsN the binding site is partially occluded by a loop connecting $\alpha 2$ with $\beta 4$ (residues 308-313). Interestingly, SPOR-FtsN presents two Cys residues (C312 and C252) in this region that could rearrange upon disulfide bond formation; nicely explaining the observed experimental correlation between PG binding and disulfide bond formation

in FtsN (Duncan et al. J Bacteriol 195, 5308-5315, (2013)). This information has been included in the main text (page 21, lines 683-686) as well as in Supplementary Figure 10.

16. Line 446. Typo: delete “collective the process”:

Done.

17. Conclusions (lines 428-479). This section is problematic on multiple levels. Only the first and last paragraphs actually pertain to insights or advances that stem from the study under review. I suggest keeping these two paragraphs and deleting the intervening two paragraphs, which summarize ideas lifted from other investigators, for most part without proper attribution. Specific concerns follow:

This section has been rewritten according to Reviewer’s suggestion.

18. Lines 447-455. These sentences rehash de Boer’s 2009 model for how PG hydrolases work together to regulate recruitment of SPOR proteins to the divisome without citing him. There is no new information in the manuscript under review that pertains to the model, and no new suggestions in the passage, so I suggest dropping it. If it is to be kept the passage needs to be revised to make it clear that these are not original insights. The basic model was first published in Gerding et al., 2009. Noteworthy extensions are in Busiek & Margolin, 2014, who showed FtsN also has a cytoplasmic localization signal that complicates the story, and in Yahashiri et al., 2015, who provided experimental support for the role of LTs in releasing SPOR proteins from septal PG. A review from Weiss’s lab that provides a more nuanced treatment of the model and discusses additional reasons why using a SPOR domain to drive localization might be beneficial (Yahashiri et al., 2017):

We would like to keep the above-mentioned passage, subsequent to its revision according to the Reviewer’s instructions. We have included the information provided by the reviewer (page 23-24, lines 560-565).

19. Lines 456-455. This passage points out that the SPOR domain of RlpA delivers the enzyme to its substrate in the septum. There is a new finding here because the authors show for the first time that the SPOR binding site is a tetrasaccharide. But the notion that the SPOR domain is a clever way to deliver RlpA to its substrate is already in Jorgenson et al., 2014, and in Yahashiri et al., 2017. The priority of these papers should be credited:

Done as requested (page 24, lines 587-592).

20. Figure 7. It should probably be noted that the cartoon in 7B is adapted from Gray et al., 2015:

This cartoon is now Figure 1B. The Gray citation is given in the legend.

The cartoon in 7C illustrating sequential activity of amidases followed by RlpA is nice, and the point that the SPOR domains binds a tetrasaccharide is an original insight, but the model was first put forth by Jorgenson et al., 2014. Their priority should be noted:

Done as requested. The citation is in the figure legend.

21. Line 466-468. This passage asserts that all previous localization studies were done in living cells and therefore it has not been possible until now to exclude that SPOR domains localize to the septum by binding a protein rather than a PG structure. This is not correct. Yahashiri et al., 2015, demonstrated localization on purified PG sacculi, which (a) is not in a living cell, and (b) pretty much ruled out the potential involvement of a septal protein:

We agree and rewrote the sentence (pages 24-25, lines 593-596).

22. Line 468-470. Lines 468-470 read “The work that we disclose herein provides the first clear direct evidence that SPOR-RlpA binds denuded PG to the exclusion of PG with peptide stems.” Claims of priority like this are always problematic because new findings build on previous work and what constitutes “clear evidence” to some might not be so unambiguous to others. In the manuscript under review, the authors document binding of a purified SPOR domain to chemically defined PG ligands, showing that a denuded glycan binds but a glycan with stem peptides does not. These experiments go beyond what has been demonstrated before. Nevertheless, I do not think the issue was up for grabs because the previous data were clear and direct enough. In particular, Yahashiri et al. showed that purified SPOR domains only bind to PG sacculi that contain denuded glycans. Their data included showing that amidase treatment of PG increased binding, while selective removal of denuded glycans with RlpA abrogated binding. If the authors think those experiments come with important caveats they should spell out the credible alternative interpretation(s) that their new findings now exclude. But I think the important new contribution here is explaining the requirement for a denuded glycan more so than showing such a requirement in the first place:

Per these comments, we rewrote the corresponding sentence (page 24, lines 589-592).

23. Line 471. Here the authors seem to be taking credit for providing evidence denuded glycans are enriched in septal PG. But priority again goes to Yahashiri et al., 2014:

We apologize if this sentence came across that way (it was not the intention). Indeed, we had/have stated the priority of this work in the first lines of the introduction section. In any case, we have rewritten it in the revised manuscript to avoid this impression clearly (page 24-25, lines 593-596).

24. Supplemental Figure 10. This figure presents models for how a glycan ligand binds to SPOR domains from CwlC, FtsN and DamX. As noted elsewhere, some of the modeled interactions have experimental support from previous site-directed mutagenesis. Although the authors allude to those studies in the general sense, somewhere in the paper they need to address the concordance between the experimental data and their models in detail:

Agreed. Please see the additional text introduced on revision (pages 21-22, lines 479-512).

Also, should DamX K414 be K413?:

Yes, this is a typo and has been corrected in the labels of Supplementary Figure 10.

REVIEWERS' COMMENTS:

Reviewer #2 (Remarks to the Author):

The authors have made enormous efforts to address all of ours, and the other's reviewers comments. While not all of it panned out (cellular localisation, for example) as anticipated, I think they have definitely done enough and this work should now be published as presented. It remains an important and well-performed study.

Reviewer #3 (Remarks to the Author):

Alcorlo et al. have made a sincere effort to address my concerns. The revised manuscript corrects errors, integrates the insights from their new structures with previous genetic studies, and documents the importance of a cluster of positively charged residues in binding glycans (what the authors call the "basic patch"). I think this work is a significant contribution to our understanding of a protein domain that is very important for bacterial cell division.

There are still a small number of mistakes or ambiguities that ought to be clarified. None of these corrects should require any further experimental work.

Table S3 is new and compares amino acids used to engage glycans in different SPOR domains. This is a welcome addition to the manuscript but appears to be incomplete and contains several mistakes.

1. In DamX Ser354 is mislabeled as 355.
2. Asterisks are used to denote residues that have been tested in vivo by mutagenesis. In FtsN Asn281 should have an asterisk. In DamX Ser354, W385, W364 and N360 should have an asterisk.
3. Some of the inferred interactions are supported by genetic studies showing substitutions at these sites reduce localization, while other inferred contacts are by this criterion not very important. The table should be revised to incorporate this information. Maybe an asterisk for the tested residues that are important and some other symbol for tested residues that are not important. The relevant data are as follows. In FtsN changing Gln251, Trp283 and R285 reduced septal localization, but changing Asn281 did not. In DamX changing Gln351, Ser354, Trp416 reduced septal localization, but changing Trp385, Trp364, or Asn360 did not.
4. Residues that interact with the glycan through the backbone instead of the R group have been omitted. Does that mean DamX Ser354 should be removed from the table? If it stays, should the corresponding residues in CwIC and FtsN be included?
5. Several amino acids in CwIC, FtsN and DamX that are depicted as contacting the glycan in the models shown in Fig S10 are not listed in the table. If this is because the table is restricted to residues with counterparts in RlpA, the title or legend should be revised to clarify this point. Otherwise, the missing amino acids should be added to the table.
6. In FtsN changing S354, Ile313 and to a lesser extent Arg256 reduced localization in vivo, so they should probably be included if the purpose of the table is to summarize "residues involved in PG binding."

There are several places where the text should be edited to improve readability or correct minor

mistakes.

7. Line 349: typo: "detriments" should be "detrimental" or "very detrimental."
8. Line 498: change "sustained" to "supported."
9. Line 508 change the semicolon ; to a comma ,
10. Line 511 change "homolog" to "homologous."
11. Line 512 change "play" to "plays."
12. Line 540: Change format of DamXspor and FtsNspor to SPOR-DamX and SPOR-FtsN to be consistent with rest of manuscript.
13. Line 540: Change "glycan" to "glycans."
14. Line 541: Change "bind" to "binds" and "which were edified by" to "as revealed by."
15. Line 542: Change to, "Our structural study reveals a wealth of new detail about the SPOR-glycan interaction"
16. Line 549: Change "they" to "it"
17. Line 569: Change "firstly" to "first"
18. Line 580: Minor correction. RlpA is not unique because a Neisseria ItgC mutant also has a chaining defect (Cloud and Dillard, 2004). Maybe say, "Pseudomonas RlpA and Neisseria LtgC are the only examples known so far of..."
19. Line 585: Change "that is" to "that are"
20. Line 596: Minor correction. The two references for demonstrating SPOR domain binding on purified sacculi should be Yahashiri et al. 2015 and Yahashiri et al. 2017. Jorgenson et al. did not do this.

Point by point answer to the reviewers comments. Please find the requested points below

REVIEWERS' COMMENTS:

Reviewer #2 (Remarks to the Author):

The authors have made enormous efforts to address all of ours, and the other's reviewers comments. While not all of it panned out (cellular localisation, for example) as anticipated, I think they have definitely done enough and this work should now be published as presented. It remains an important and well-performed study.

We are grateful to the Reviewer#2 for her/his positive comments. We are pleased to see that our efforts have met her/his approval.

Reviewer #3 (Remarks to the Author):

Alcorlo *et al.* have made a sincere effort to address my concerns. The revised manuscript corrects errors, integrates the insights from their new structures with previous genetic studies, and documents the importance of a cluster of positively charged residues in binding glycans (what the authors call the “basic patch”). I think this work is a significant contribution to our understanding of a protein domain that is very important for bacterial cell division.

We are grateful to the Reviewer#3 for her/his positive comments.

There are still a small number of mistakes or ambiguities that ought to be clarified. None of these corrects should require any further experimental work.

As detailed below, all these points have been corrected/clarified in the revised version.

Table S3 is new and compares amino acids used to engage glycans in different SPOR domains. This is a welcome addition to the manuscript but appears to be incomplete and contains several mistakes.

1. In DamX Ser354 is mislabeled as 355.

Corrected.

2. Asterisks are used to denote residues that have been tested *in vivo* by mutagenesis. In FtsN Asn281 should have an asterisk. In DamX Ser354, W385, W364 and N360 should have an asterisk.

Done as suggested.

3. Some of the inferred interactions are supported by genetic studies showing substitutions at these sites reduce localization, while other inferred contacts are by this criterion not very important. The table should be revised to incorporate this information. Maybe an asterisk for the tested residues that are important and some other symbol for tested residues that are not important. The relevant data are as

follows. In FtsN changing Gln251, Trp283 and R285 reduced septal localization, but changing Asn281 did not. In DamX changing Gln351, Ser354, Trp416 reduced septal localization, but changing Trp385, Trp364, or Asn360 did not.

The Supplementary Table 3 has been modified according to reviewer's request.

4. Residues that interact with the glycan through the backbone instead of the R group have been omitted. Does that mean DamX Ser354 should be removed from the table? If it stays, should the corresponding residues in CwlC and FtsN be included?

We decided to include in Table S3 the corresponding residues in CwlC and FtsN (Ala192 and Ser254 respectively).

5. Several amino acids in CwlC, FtsN and DamX that are depicted as contacting the glycan in the models shown in Fig S10 are not listed in the table. If this is because the table is restricted to residues with counterparts in RlpA, the title or legend should be revised to clarify this point. Otherwise, the missing amino acids should be added to the table.

To address this point, we have modified the title of the Supplementary Table 3 that now stands: "Residues with counterparts in RlpA involved in PG binding among different SPOR domains."

6. In FtsN changing S354, Ile313 and to a lesser extent Arg256 reduced localization in vivo, so they should probably be included if the purpose of the table is to summarize "residues involved in PG binding."

We guess the reviewer is referring to S254. As suggested, we have modified Supplementary Table 3 accordingly.

There are several places where the text should be edited to improve readability or correct minor mistakes.

7. Line 349: typo: "detriments" should be "detrimental" or "very detrimental."

Done

8. Line 498: change "sustained" to "supported."

Done

9. Line 508 change the semicolon ; to a comma ,

Done

10. Line 511 change "homolog" to "homologous."

Done

11. Line 512 change "play" to "plays."

Done

12. Line 540: Change format of DamXspor and FtsNspor to SPOR-DamX and SPOR-FtsN to be consistent with rest of manuscript.

Done

13. Line 540: Change “glycan” to “glycans.”

Done

14. Line 541: Change “bind” to “binds” and “which were edified by” to “as revealed by.”

Done

15. Line 542: Change to, “Our structural study reveals a wealth of new detail about the SPOR-glycan interaction”

Done

16. Line 549: Change “they” to “it”

Done

17. Line 569: Change “firstly” to “first”

Done

18. Line 580: Minor correction. RlpA is not unique because a *Neisseria* ItgC mutant also has a chaining defect (Cloud and Dillard, 2004). Maybe say, “*Pseudomonas* RlpA and *Neisseria* LtgC are the only examples known so far of...”

Done

19. Line 585: Change “that is” to “that are”

Done

20. Line 596: Minor correction. The two references for demonstrating SPOR domain binding on purified sacculi should be Yahashiri et al. 2015 and Yahashiri et al. 2017. Jorgenson *et al.* did not do this.

Done